# Advancements in Oncoproteomics Technologies: Treading toward Translation into Clinical Practice

**DOI:** 10.3390/proteomes11010002

**Published:** 2023-01-10

**Authors:** Ankita Punetha, Deepak Kotiya

**Affiliations:** 1Department of Microbiology, Biochemistry and Molecular Genetics, Rutgers New Jersey Medical School, Rutgers University, 225 Warren St., Newark, NJ 07103, USA; 2Department of Pharmacology and Nutritional Sciences, University of Kentucky, 900 South Limestone St., Lexington, KY 40536, USA

**Keywords:** proteoform, oncoproteomics, prognostic and diagnostic biomarker, mass spectrometry, protein microarray, tissue microarray, antibody microarray, cancer, drug discovery

## Abstract

Proteomics continues to forge significant strides in the discovery of essential biological processes, uncovering valuable information on the identity, global protein abundance, protein modifications, proteoform levels, and signal transduction pathways. Cancer is a complicated and heterogeneous disease, and the onset and progression involve multiple dysregulated proteoforms and their downstream signaling pathways. These are modulated by various factors such as molecular, genetic, tissue, cellular, ethnic/racial, socioeconomic status, environmental, and demographic differences that vary with time. The knowledge of cancer has improved the treatment and clinical management; however, the survival rates have not increased significantly, and cancer remains a major cause of mortality. Oncoproteomics studies help to develop and validate proteomics technologies for routine application in clinical laboratories for (1) diagnostic and prognostic categorization of cancer, (2) real-time monitoring of treatment, (3) assessing drug efficacy and toxicity, (4) therapeutic modulations based on the changes with prognosis and drug resistance, and (5) personalized medication. Investigation of tumor-specific proteomic profiles in conjunction with healthy controls provides crucial information in mechanistic studies on tumorigenesis, metastasis, and drug resistance. This review provides an overview of proteomics technologies that assist the discovery of novel drug targets, biomarkers for early detection, surveillance, prognosis, drug monitoring, and tailoring therapy to the cancer patient. The information gained from such technologies has drastically improved cancer research. We further provide exemplars from recent oncoproteomics applications in the discovery of biomarkers in various cancers, drug discovery, and clinical treatment. Overall, the future of oncoproteomics holds enormous potential for translating technologies from the bench to the bedside.

## 1. Introduction

Proteomics is the study of the proteome. The proteome encompasses the entire set of proteoforms present at a certain time in a cell, tissue, or individual in a given biological setting. Proteomics includes the assessment of global protein abundance, proteoform levels, spatial conformations, chemical modifications, cellular localization, proteoform functions, cofactors, and interacting partner networks. In the field of proteomics, there has been a paradigm change from protein expression to proteoform abundance from the genome [1,2]. The variations in protein product may result from genetic changes, mutations, transcriptional variations, RNA splicing, translational error, protein folding, proteolytic cleavage of a signal peptide, or a myriad of post-translational modifications (PTMs). This yields a variety of protein products relative to the canonical form. These diverse molecular forms of a protein product of a single gene are termed ‘proteoforms’ [1,3]. Each proteoform has a specific subcellular location, where it interacts with surrounding molecules and may form a complex to carry out a specific biological function, and consequently have important effects at the system level [1,3,4]. Proteoforms can, therefore, act as the ultimate long-range functional effectors of a gene and increase the structural and functional diversity of the proteome. Further, the extensive temporal dynamic range of the proteoforms adds complexity to proteome analysis. Innovative and progressive proteomic technologies are needed for large-scale analysis of these broad-range processes. The physiological and pathological processes may have a varying abundance of a particular proteoform and may exhibit changes in localization or response to stimuli, which makes them highly relevant to intervention and drug discovery in various diseases [5,6]. For instance, the five clinical areas of interest where proteoforms are linked to the progression of diseases include (1) neurodegeneration (e.g., the hyperphosphorylation of Tau results in Alzheimer’s disease), (2) cardiovascular disease (e.g., the phosphorylation of cTnl results in cardiac injury), (3) infectious diseases (e.g., glycerophosphorylation of PilE results in cerebrospinal meningitis), (4) immunobiology (e.g., glycosylation of a monoclonal antibody is used in antibody-based drugs and diagnosis), and (5) cancer (e.g., hypervariation in KRAS4B results in tumor-specific proteoforms) [5]. A comprehensive knowledge of proteoform structure and properties will, therefore, help in deciphering its function in basic and translation research.

Cancer is a leading cause of death worldwide which resulted in nearly 10 million deaths in the year 2020 [7,8]. Based on 2019 CDC incidence data, there were 1,752,735 new cases of cancer and 599,589 cancer-associated deaths [9]. The incidences of cancer and its associated mortality are increasing globally, which calls for more effective and sensitive sets of biomarkers for an early diagnosis and consecutive intervention [9]. Cancer is a large group of diseases that can affect any part of the body. The cancerous cells divide uncontrollably and grow beyond their usual boundaries to invade adjoining tissues/organs and exhibit metastasis. The complications arising from widespread metastasis to neighboring, as well as distant, organs are the major cause of death from cancer. Proteoforms play a critical role in tumorigenesis. They relay cellular information, provide structure, repair DNA damage, maintain cellular metabolism, cell cycle, and apoptosis. To perform these tasks, they may form supra-molecular assemblies and any change in proteoform structure, abundance, or interactors will impact cellular function. The pathology of cancer development involves multiple environmental carcinogenic factors and genetic alterations that change the proteome in complex ways that is way beyond simple alterations in proteoform abundance. Cancer is a complex and heterogeneous disease [10] and the onset and progression involves multiple dysregulated proteoforms and their downstream signaling pathways [11]. Usually, the mutations in tumor suppressor genes or oncogenes result in the formation of aberrant proteoforms, which disrupt signaling pathways, impair cellular function, and eventually lead to cancer.

Oncoproteomics comprises the systematic study of proteins including various proteoforms and their interactions in cancer using proteomics technologies. It helps to identify and quantify proteoforms abundance, changes in PTMs patterns, and interaction networks between the healthy and diseased tissue at different stages from preneoplasia to neoplasia. The information is utilized to evaluate cancer prognosis, diagnosis, tumor classification, develop cancer therapeutics, and distinguish potential responders for particular therapies [12,13,14,15], thus increasing the understanding of cancer pathological mechanisms. Additionally, proteomics has been applied to investigate the alteration in the signaling pathways in tumor cells, providing insight to tweak numerous pathways for cancer therapies. The individualized selection of therapeutic combinations will help in targeting the entire cancer-specific protein network. With the advent of advanced technologies, the therapeutic efficacy and toxicity can be now monitored in real-time, facilitating the modulation of therapies based on the changes in the specific protein network with cancer prognosis and drug resistance [16,17,18,19,20,21]. The creation of cancer proteome databases that contain a huge amount of proteomics data, protein interactome, integrated with cancer genomics data, and clinical information is greatly benefitting the analysis. Thus, oncoproteomics technologies help to interrogate the proteome to discover novel biomarker candidates for early diagnosis and prognosis of cancer, its surveillance, identify novel therapeutic drug targets, develop new drugs and targeted molecular therapies, study drug efficacy and toxicity, monitor treatment in real-time, and manage personalized cancer medication [16]. These technologies are being developed for routine application in clinical settings.

The protein sources can be cell lines, tumor tissue, or body fluids such as blood, serum, and urine (Figure 1). In a typical pipeline of proteome analysis, the extracted or purified protein products can be either fractionated directly (a top-down approach) or after protease (usually tryptic) digestion (a bottom-up approach), and analyzed using mass spectrometry (MS) to identify proteins, and the data can be interpreted using a proteome database [22,23,24,25,26]. In clinical research, label-based and label-free MS approaches are utilized for quantitative analysis [27,28,29,30]. Multiplex and innovative technologies like protein-, antibody-, tissue microarray, proximity extension assay, nanoproteomics and single-cell proteomics have significantly improved protein purification and automation in the identification of protein traces in minuscule samples [31,32,33]. Thus, proteomics facilitates the concurrent qualitative and quantitative profiling of several proteoforms that allows the discovery of sensitive and specific cancer biomarkers [28,33]. There are various interesting reviews available in the literature that discuss specific aspects of proteomics in cancer [16,32,33,34,35,36,37], such as protein biomarker discovery [16], MS-based clinical proteomics [33], or enrichments of PTMs [35], to name a few. However, currently, there are no reviews that cover the wide range of techniques that are being utilized in oncoproteomics. This review is intended to fulfill the lacuna and provide numerous techniques including the contemporary approaches that have reached clinical settings to unravel cancer biology. The approaches have been described in a way to aid the understanding of a broad range of readers. Further, to appreciate the enormous potential of proteomics technologies exemplars from recent oncoproteomics applications in the biomarker discovery in various cancers, drug discovery, and clinical treatment are provided.

## 2. Advances in Proteomic Technologies Used in the Study of Cancer

This section discusses the technologies used in proteomic profiling and the investigation of diagnostic and prognostic biomarkers in different cancers.

### 2.1. Gel-Based Approaches

Gel-based approaches are versatile methods of global protein separation that are discussed in the following section.

#### 2.1.1. Two-Dimensional Gel Electrophoresis

Two-dimensional gel electrophoresis (2-DE) is an important and well-established technical platform for the reliable and efficient separation of proteins based on relative mass (Mr) and charge [38]. The conventional concepts of 2-DE combine isoelectric focusing (pI) with sodium dodecyl sulfate-polyacrylamide gel electrophoresis (SDS-PAGE), which results in the resolution of thousands of spots in one gel. High-resolution 2-DE can even resolve up to 10,000 protein spots (including separation of proteoforms) per gel [22,39]. The developments in proteomics and proteome enrichment have revealed that each protein contains a series of proteoforms, as discussed earlier in the introduction. The proteoforms arising from a single gene with a specific Mr and pI are expected to be distributed in different 2-DE spots and, therefore, can be resolved by 2-DE [40,41]. The changes in Mr and pI are recognized by the horizontal or vertical shifting of a spot. However, each 2-DE spot may contain a few to several hundred proteoforms derived from different genes [41], as proteoforms with very similar Mr and pI in proteomes can comigrate into a spot. Different proteoforms in a single spot usually have significant abundance differences. The low-abundance proteoforms can be identified and quantified using high-resolution mass spectrometers. The difference in intensity and position of gel spots can be compared between disease versus healthy controls to identify changes in proteoforms abundance or any chemical alterations in proteoforms [42].

The resolved proteoforms in the gel are visualized by stains, such as Coomassie brilliant blue or silver staining, and spot intensities between gels can be analyzed. The abundance of different proteoforms can be identified by a direct side-by-side comparison of gels from different sample states [43]. The major disadvantages of 2-DE are low dynamic range, extensive labor, and gel-to-gel variability that may hinder the comparison and relative quantification of spots from different 2-DE experiments.

#### 2.1.2. 2D Differential in-Gel Electrophoresis

The 2D differential in-gel electrophoresis (2D DIGE) allows parallel comparison of multiple protein samples within the same gel, thus facilitating relative comparison of different sample states without gel-to-gel variability. For quantitative analysis by DIGE, the protein samples are labeled with spectrally distinct, charge- and mass-matched fluorescence dyes, such as Cy2, Cy3, or Cy5, and mixed before electrophoresis and run along with a differently labeled standard on the same 2D gel. The individual protein products are differentially visualized by differential fluorescence [44,45]. For identification, the gel-separated protein products can be either probed with antibodies, or digested into peptides to obtain a peptide mass fingerprint that can be examined against theoretical fingerprints of protein sequences in the database [46], or subjected to high-resolution mass spectrometry (MS) for accurate mass determination [47,48]. Moreover, 2-DE has been used in proteome analysis of human tissue, plasma, and serum, with or without prior fractionation [48,49,50]. DIGE has improved accuracy over 2-DE and can be utilized in biomarker discovery that does not involve high-throughput sample processing [51,52,53].

### 2.2. Mass Spectrometry-Based Approaches

Mass spectrometry (MS) is an analytical tool that is used to measure the mass-to-charge ratio (*m/z*) of one or more molecules present in a sample. The results are obtained as a mass spectrum, which is a plot of ion signal (the intensity) as a function of the *m/z* ratio. These spectra are used to determine the elemental or isotopic signature, exact molecular masses of the sample components, and elucidate the chemical identity or structure of molecules. Thus, MS can be used to (1) identify unknown compounds by determining molecular weight, (2) quantify known compounds, and (3) elucidate the structure and chemical properties of molecules [54,55]. Moreover, it can be applied to pure samples as well as complex mixtures.

A mass spectrometer typically consists of three major components: an ion source, a mass analyzer, and a detector. In a typical MS procedure, the sample whether solid, liquid, or gaseous is first ionized, and magnetic and/or electric fields are used to separate ions by virtue of their different trajectories (based on their *m/z* ratio) in a vacuum that is finally detected by the detector. Ion source: Each phase (solid, liquid, or gaseous) requires different ionization methods. The ionization may be continuous or pulsed and may occur at different pressures. The ions generated may be positively or negatively charged. In biomedical applications, samples are predominantly liquids containing large molecules that require continuous soft ionization to avoid fragmentation, such as electrospray ionization (ESI) or matrix-assisted laser desorption/ionization (MALDI). Mass analyzer: The mass analyzers can use magnetic and/or electric fields with a static or time-varying field, and operation is made continuous or cyclic. The main variants of mass analyzers include the magnetic sector, Fourier-transform ion cyclotron resonance, quadrupole, ion trap, and time-of-flight (TOF) mass spectrometer. Usually, electric fields are preferred because they avoid the requirement of a large, heavy magnet. Subsequently, the quadrupole, ion trap and TOF mass spectrometer are preferred [56]. All offer high performance with several advantages, such as sensitivity, mass resolution, and mass range based on the requirement. Sensitivity limits are set by the ion flux and space charge effects at low and high fluxes; the mass resolution is limited by the thermal spread in ion velocity and the precision of the applied fields, while the mass range is limited by the magnitude of the field. Detector: The final element of the MS is a detector that records the charge induced or the current produced when an ion passes by or hits a surface.

MS can adopt various forms such as deep-discovery MS (e.g., LC-MS/MALDI) or targeted/directed (e.g., single reaction monitoring (SRM), multiple reaction monitoring (MRM), or parallel reaction monitoring (PRM)). However, for a successful clinical implementation, a streamlined workflow is indispensable. The widely adopted pipeline includes de novo MS discovery followed by low- or medium-plex targeted MS for downstream analysis. The pipelines intended for biomarker profiling in clinical settings usually include deep-discovery MS followed by targeted MS and then high-resolution MS [57,58,59].

The MS approaches can provide qualitative, semi-quantitative, or quantitative data. The quantification methods encompass label-based (e.g., isotope labeling including metabolic or chemical labeling) or label-free (e.g., emPAI) approaches, which will be discussed in upcoming sections.

#### 2.2.1. Liquid Chromatography–Mass Spectrometry

In liquid chromatography–mass spectrometry (LC-MS) systems, the liquid analytes are first separated by LC and individual molecules are passed sequentially into the mass spectrometer to identify their masses. LC variants with an increase in pressure include high-performance LC (HPLC) and ultra-performance LC (UPLC). The LC column effluent is nebulized, desolvated, and softly ionized using ESI, creating charged particles. In ESI, the solubilized sample is passed through a high-voltage needle held at atmospheric pressure that produces charged droplets, which destabilize and explode into finer droplets. The desolvated analyte ions migrate under a high vacuum through a mass analyzer that separates ions based on their *m/z* ratio and transfers them into a detector. LC-MS instrument is usually an HPLC unit with an attached mass spectrometer and LC-MS/MS is an HPLC with two mass spectrometers.

Tandem MS (*MS*/*MS)* consists of two mass analyzers that have been shown to improve speed and sensitivity and are used for the analysis of protein or peptide mixtures or the determination of the mass of intact protein product. It is commonly used for proteome analysis of complex biological samples (such as human serum or feces) where the overlap between peptide masses cannot be resolved with a high-resolution mass spectrometer. The first mass analyzer is used to isolate the precursor ions that are subsequently fragmented in a collision cell. The resulting fragment ions are then separated in the second mass spectrometer, generating a pattern of fragments (the tandem mass spectrum), which forms the characteristic fingerprint of the molecule of interest. The most popular mass analyzers used in tandem MS include quadrupole (Q), time-of-flight (TOF), or hybrid analyzers, such as quadrupole coupled with TOF (Q-TOF), depending on the data required (structural or quantitative), resolution, and mass accuracy.

The quadrupole mass analyzer consists of four parallel cylindrical metal rods at a well-defined distance from each other. A combination of direct current (DC) and radio frequency (RF) voltages is then applied to the rods, creating a time-varying quadrupolar field that separates ions based on the stability of their trajectories. At a particular ratio of DC to RF voltage, ions with specific *m/z* will have confined trajectories and without discharging will pass through the length of the quadrupole. The disadvantage of quadrupole is that length, constructional precision, the frequency of the RF voltage limits its mass selectivity, and the amplitude of the RF voltage limits mass range [56,60].

The TOF mass analyzer provides high mass accuracy and range. In the ion modulator region of the TOF analyzer, ions are accelerated under an electric field to acquire similar kinetic energy and then admitted to a field-free drift region of the flight tube for mass separation. Ions become separated based on their *m/z* value by measuring the time taken to traverse a known distance before striking a detector. The lighter ions travel faster and the heavier ions take longer to travel, as the square of the drift time of an ion is proportional to its *m*/*z* ratio. A mass spectrum is generated, representing the number of ions hitting the detector over time. A full mass spectrum can be obtained by scans of the whole mass range, which enables the determination of the molecular masses of the ions with high accuracy. High mass range/resolution can be obtained by short pulse, low axial velocity, and large distance (the length of the flight path). However thermal energy causes uncertainty in the initial position and velocity of the ions which can be optimized by orthogonal acceleration, delayed ion extraction, or using a reflectron (reflection by a stacked electrode) to reach a much higher resolution than linear TOF. For instance, in the case of reflectron TOF, a contrary electric field is applied at the end of the TOF tube to push the ions back at a single angle from the original axial direction. This corrects kinetic energy dispersion and spatial spread of ions that exhibit the same *m/z* but different velocities, which allows ions of the same *m/z* to arrive at the detector at the same time. The addition of a reflectron also increases the flight path length which improves mass resolution [55,56]. To improve resolution further, Hadamard transform or tandem TOF can be used [55,56,61]. Hadamard transform mass analysis significantly increases the signal-to-noise ratio and several ions traveling in the flight tube can be analyzed simultaneously [61]. In tandem TOF, two TOF analyzers are used consecutively [62,63]. The first TOF analyzer consists of a flight tube with the timed ion selector isolating the precursor ions of choice using a velocity filter, while the second TOF analyzer typically contains a post accelerator, flight tube, and ion mirror. The ion detector analyzes the fragment ions.

Q-TOF MS is a hybrid mass analyzer that advantageously combines the ion selection properties of a quadrupole with the high speed, mass resolution, and accuracy of a TOF in a single system. It usually has two quadrupole systems and a TOF tube. The first quadrupole acts as a mass filter for the selection of specific ions based on their *m/z* ratio and the second quadrupole acts as a collision cell where ions are bombarded by inert gas molecules, such as nitrogen or argon, resulting in the fragmentation of the ions by a process known as collision-induced dissociation (CID). In wide band pass/RF only mode, there is no gas in the collision cell and all ions from the quadrupole are transferred into the TOF analyzer without subsequent fragmentation of ions. However, in narrow pass mode, fragmentation of a selected ion with a known *m/z* value occurs and the quadrupole acts as a filter to pass ions with a particular *m/z* value into the TOF analyzer. Most ions produce a signature fragmentation pattern that can be identified using databases or chemical standards. The ions with the same mass can be differentiated based on their fragmentation pattern. The Q-TOF offers high mass accuracy together with tandem MS, which is suitable for nontargeted profiling applications [64,65].

#### 2.2.2. Matrix-Assisted Laser Desorption/Ionization

Matrix-assisted laser desorption/ionization (MALDI) is a soft ionization technique used in MS that involves a laser collision with a matrix of molecules to make the analyte molecules into the gas phase without fragmenting or decomposing them. It is suitable for identification and spatial distribution studies of large biomolecules, which are either non-volatile or thermally unstable. The analyte dissolved in a solvent containing a selected matrix, such as sinapic acid or α-cyano-4-hydroxycinnamic acid, is deposited on a target plate for drying and crystallization. In a variant of MALDI called SALDI (surface-assisted laser desorption/ionization), the solid nanomaterial is used as the matrix, which provides a more homogeneous sample distribution. The target plate is then placed in the vacuum chamber of a mass spectrometer and bombarded by photons from a pulsed laser, resulting in the desorption and ionization of the matrix. The energy from the matrix is gently transferred to the sample molecules leaving it intact and in the gas phase, yielding protonated (cationized) or deprotonated (anionized) molecular ions. The ions are then separated based on their TOF which is proportional to their *m/z* value.

The ability to desorb large molecules, high accuracy, high sensitivity, and wide mass range makes MALDI-TOF MS a method of choice in clinical settings for the identification of biomolecules in complexes [66,67] and cancer diagnosis and prognosis [68]. For instance, MALDI-TOF MS is used in differentiating ovarian cancer from healthy controls. One of the two approved biomarkers by the FDA for the diagnosis of recurrence and treatment response in ovarian cancer is the Cancer antigen 125 (CA125), which predicts cancer up to 9 months before diagnosis. However, CA125 is not produced by early-stage tumors and might be controlled by other benign gynecological diseases. The combined use of CA125 with two MALDI-TOF MS feature peaks, detected as connective tissue-activating peptide III and platelet factor 4, resulted in earlier detection of ovarian cancer than using CA125 alone [69]. Rapid screening tools for early-stage ovarian cancer detection based on MALDI-TOF MS of blood serum have been developed [70]. The specific sample pre-treatment methods can improve the MALDI-TOF MS-based diagnosis of ovarian cancer. For instance, the analysis of ions with low mass from serum after the removal of high-level proteoforms performed better in differentiating diseased from healthy controls [71]. Similarly, before MALDI-TOF MS analysis, the extraction of low-abundance proteoforms via enrichment technology, such as magnetic beads, can be performed [72]. A solid-phase extraction before analysis has been shown to improve the sensitivity in diagnosis to differentiate serous adenocarcinoma (a common type of epithelial ovarian cancer) from healthy controls [73]. The combination of iTRAQ-based quantitative proteome analysis and MALDI-TOF MS has been proposed to perform better in differentiating benign and malignant tumors in ovarian cancer [74]. MALDI-TOF MS has been used in the early diagnosis and prognosis of various other cancers, such as prostate cancer [72,75], liver cancer [76], and multiple myeloma [77,78]. Furthermore, the combination of two types of spectroscopic techniques, surface-enhanced Raman scattering (SERS) and MALDI-TOF MS in plasma exosome profiling, has been shown to rapidly differentiate osteosarcoma patients from healthy controls with higher precision than either technique [78].

#### 2.2.3. MALDI Mass Spectrometry Imaging

MALDI mass spectrometry imaging (MALDI-MSI) is a powerful technique by which the spatial and temporal distribution of proteoforms and biomolecules can be investigated directly from a tissue section without the need for extraction, purification, and separation measures [79]. The MSI is based on the mapping of the corresponding ion intensities along with the determination of the spatial distribution of many molecules in a sample. In MALDI-MSI, a tissue section is coated with matrix and the sample is raster-scanned (with a spatial resolution ranging from approximately 200 μm down to 20 μm) in the mass spectrometer resulting in spatially resolved mass spectra. The laser only strikes the matrix crystals without affecting the tissue section. After the MALDI measurement, histological staining allows a histology-directed analysis of the mass spectra. To reduce analyte diffusion which alters the original distribution and reduces the spatial resolution, matrix-free ionization platforms have been developed for use, such as inorganic matrix and nanophotonic platforms instead of organic matrices [80].

MALDI-MSI has the advantage of correlating the MALDI images (molecular information) with histological information by keeping the spatial localization information of the analytes after the MS measurement. MALDI-MSI software is used to superimpose the MALDI images over an optical image of the sample. The MSI data analysis tools include integrated open-source software packages, such as MSIReader [81], OmniSpec [82], Cardinal [83], or the freely available open MSI platforms [84,85], BioMap (Novartis), DataCubeExplorer (AMOLF), Mirion (JLU) [86], SpectViewer (CEA), or the integrated commercial packages from instrument manufacturers, such as Xcalibur/ImageQuest (Thermo Fisher Scientific), SCiLS (Bruker Daltonics), High Definition Imaging (HDI, Waters Corporation), and BASIS [87]. The details of MALDI-MSI are covered in other reviews [32,88,89,90].

MALDI-MSI is a label-free technique that allows multiplex analysis of numerous molecules in the same tissue section and has been extensively employed in clinical proteomics in cancer [91,92,93]. It has been used to understand primary pancreatic ductal adenocarcinoma and metastases [92], molecular signatures of medullary thyroid carcinoma [93], tumor-stroma interrelationships [94], de novo discovery of phenotypic intratumor heterogeneity [95], and as a complementary diagnostic tool in cytopathology for thyroid nodules characterization [96]. Recently, MALDI imaging along with deep learning has been employed in multi-class cancer subtyping in salivary gland carcinomas [97]. In translational MSI, to improve transparent and reproducible data analysis, implementation of the Galaxy framework has been shown for the urothelial carcinoma dataset [98].

#### 2.2.4. Surface-Enhanced Laser Desorption/Ionization Time-Of-Flight Mass Spectrometry

The Surface-enhanced laser desorption/ionization time-of-flight mass spectrometry (SELDI-TOF-MS) technique is one variant of MALDI that uses surface extraction by ProteinChip. It was introduced in 1993 [99] and later commercialized as the ProteinChip system by Ciphergen Biosystems in 1997. In SELDI, the sample containing the protein mixture is applied on a surface customized with a chemical functionality, such as binding affinity. The different substances used to modify SELDI surface may include antibodies, receptors, ligands, nucleic acids, carbohydrates, metal ions, or chromatographic surfaces (e.g., cationic, anionic, hydrophobic, or hydrophilic exchangers). The protein product of interest becomes sequestered by interacting with substances on the surface of ProteinChip based on biological or chemical affinities. The nonspecific substances and contaminants are removed by subsequent on-spot washing and only the surface-bound protein products are left for analysis. An energy absorbing matrix (e.g., SPA) is applied to the surface for crystallization with the target molecules and then subjected to laser ionization, which delivers higher specificity and sensitivity in subsequent analysis.

Samples spotted on a SELDI surface derivatized with chromatographic functionality (e.g., normal-phase, reversed-phase, ion exchange, metal, or biological affinity) are usually analyzed with TOF-MS [100]. The advantage of this technology is the integration of on-chip selective and sensitive capture, partial characterization of the analyte, and relative quantitation. SELDI-TOF MS has been employed for the diagnosis, detection, and identification of biomarker candidates for various cancer types, such as prostate [101], pancreatic [102], lung [103], breast [104], melanoma [105], colon [106], oral squamous cell carcinoma [107], gastric [108,109], ovarian [110,111], liver [112], renal [113], and esophageal [114,115].

#### 2.2.5. Targeted/Directed Mass Spectrometry

The characteristic of targeted MS is the selection and fragmentation of a predetermined set of precursor ions that are either predicted or identified in a survey scan. The multiple stages of tandem MS are utilized for two or three ions of a specific mass at a specific time. The *m/z* values and time can be defined in an inclusion list that is derived from a previous analysis. The specific targeting of analyte peptides of interest provides exquisite specificity and sensitivity. In contrast to discovery MS, protein identification based on fragment ion spectra and protein quantification based on survey scans is decoupled and performed in two separate experiments [116,117,118]. Targeted MS is manifold sensitive than discovery MS and provides highly precise quantification using internal standards [118]. Targeted MS proteomics approaches are discussed below.

##### Single Reaction Monitoring and Parallel Reaction Monitoring-Mass Spectrometry

Single Reaction Monitoring Mass Spectrometry (SRM-MS) is a nonscanning MS technique used to selectively detect and quantify peptides based on the screening of specified precursor-to-fragment ion transitions. It is performed on triple quadrupole instruments (QQQ-MS) with CID to increase selectivity. The instrument is capable of selectively isolating the precursor ions corresponding to the *m/z* of the signature peptides and selectively monitors peptide-specific fragment ions [119]. In the first stage (mass selection), a precursor ion is selected that undergoes fragmentation to generate fragment (product) ions. The second stage involves fragment ion selection. To monitor a particular fragment ion of a selected precursor ion, the two mass analyzers are utilized as static mass filters. Instead of recording mass spectra, the detector serves as a counting device for the ions matching the selected transition and yields an intensity value over time [120].

In multiple reaction monitoring mass spectrometry (MRM-MS), multiple SRM transitions are determined within the same experiment on the chromatographic time scale by alternating between the different precursor-to-fragment ion pairs. The QQQ instrument cycles through a series of transitions and records the signal of each transition as a function of the retention time [121,122]. The examination of the chromatographic coelution of multiple transitions for a given analyte offers additional selectivity [123].

The use of SRM/MRM MS in preclinical studies and clinical laboratories facilitates rapid screening and measurement of large numbers of candidate proteins in complex biological samples for biomarker verification, which circumvents the necessity for large panels of validated antibodies [124,125,126].

##### High-Pressure and High-Resolution Separations Coupled with Intelligent Selection and Multiplexing

High-pressure and high-resolution separations coupled with intelligent selection and multiplexing (PRISM) is an antibody-free strategy that includes high-pressure and high-resolution separations coupled with intelligent selection and multiplexing for sensitive SRM-based targeted protein quantification. It utilizes high-resolution reversed-phase LC separations for analyte enrichment, SRM monitoring of internal standards-based intelligent selection of target fractions, followed by fraction multiplexing and quantification with nano LC-SRM [127,128]. It provides higher sensitivity in targeted protein quantification without any specific-affinity reagents. The method has yielded accurate and reproducible quantification of proteins from biological samples at a concentration in the pg/mL range [127,129]. PRISM-SRM has the disadvantage of reduced analytical throughput due to fractionation. However, even with limited fraction concatenation, moderate throughput can be achieved by combining fractions into fewer multiplexed fractions based on peptide elution times [128,130].

##### Parallel Reaction Monitoring

Parallel reaction monitoring (PRM) uses quadrupole equipped with high-resolution and accurate mass instruments [131,132]. The PRM uses Q Exactive to avoid lengthy assay development. A PRM instrument is like QQQ, in which the third quadrupole is replaced with a high resolution, high mass accuracy Orbitrap mass analyzer. In contrast to SRM, where specific transitions are monitored one at a time, the Q Exactive allows parallel detection of all transitions in a single analysis. Because all transitions can be monitored with PRM, it circumvents laborious optimizations to generate idealized assays for selected transitions [133]. The monitoring of all potential product ions to confirm the identity of the peptide instead of just 3–5 transitions add additional specificity [134,135].

The advantage of PRM is that it does not require prior knowledge to preselect target transitions before analysis, as it monitors all transitions. Additionally, since many ions are available, the presence of interfering ions in a full mass spectrum instead of a narrow mass range becomes less problematic to overall spectral quality [132,136]. The Q Exactive instrument is very flexible and can be deployed for both discovery and targeted analysis. This permits the combination of a discovery-based approach to identify proteins of interest followed by targeted approach to monitor targets with high sensitivity under various conditions in a single experiment [131,132,135].

##### Sequential Window Acquisition of All Theoretical Fragmentation Spectra

Sequential window acquisition of all theoretical fragmentation spectra (SWATH) is a novel technique in which data-independent acquisition (DIA) is coupled with peptide spectral library matching. It complements traditional discovery MS-based proteomics techniques and SRM/MRM methods. It performs directed label-free quantification in an SRM/MRM-like manner, with higher accuracy and precision [137,138]. The systematic queries of sample sets are made for the presence and quantity of any protein product of interest. The DIA method generated fragment ion maps are mined based on the information present in the fragment ion spectral libraries.

In SWATH acquisition, the first quadrupole sequentially cycles the precursor isolation windows, called swaths, across the mass range of interest and generates time-resolved fragment ion spectra for all the detectable analytes [137,138,139]. In this many SRM/MRM-like experiments can be performed simultaneously. In the SWATH-MS, faster acquisition speed is needed to obtain an adequate number of data points across the chromatographic peak such that ion spectra can be reconstructed with an acceptable signal-to-noise ratio. A disadvantage of SWATH acquisition is that the data is incompatible with conventional database searching and requires deconvolution algorithms to process complex data [137,138]. However, in spite of this, SWATH-MS-based quantitative proteomics is a widely adopted approach in oncoproteomics analysis [140,141,142,143,144,145,146,147].

#### 2.2.6. Quantitative Analysis Methods

The most attractive part of proteomics is its ability to reveal novel biomarkers of cancer. With the progression of cancer, changes in proteoforms and their differential distribution both in tissues and body fluids can be monitored via concurrent qualitative and quantitative profiling of numerous proteoforms. Accurate quantitation is crucial for oncoproteomics analysis. For quantitative investigations in clinical research, label-based and label-free approaches are widely used. In the case of label-based approaches, isotopic labeling is used, which involves in vivo or in vitro incorporation of stable isotopes into proteins or peptides for comparative analysis with isotope-free markers. Labeling allows multiplexing that permits simultaneous analysis of several samples and reduces experimental variability inherent in sample processing.

##### Stable Isotope Labeling by Amino Acids in Cell Culture (SILAC)

Stable isotope labeling by amino acids in cell culture (SILAC), also called metabolic labeling, involves in vivo incorporation of stable isotopes (such as deuterium, ^13^C, ^15^N, etc.), into the proteome during cell growth. The metabolically active cells are cultured with media containing isotopically labeled amino acids, particularly arginine and lysine. This is suitable for whole proteome labeling of live cells, followed by tryptic digestion and quantification by MS. SILAC is used in cell culture because of its simplicity and robustness to encode cell populations with quantifiable labels. Cell populations grown with differently labeled amino acids can be analyzed simultaneously [148,149,150]. SILAC has been adapted for use in model organisms such as mice [151,152], zebrafish [153,154], newts [155], worms [156,157], and yeast [158]. The primary limitation of SILAC is the requirement for cells to be metabolically active (i.e., undergoing active protein synthesis) so that they can incorporate the labeled amino acid. As such, it cannot be used for nondividing cells and human samples, and is expensive to apply in small mammals. This makes it difficult to use original technology for many clinical specimens. Recently, various SILAC variants have been developed, such as NeuCode SILAC, super-SILAC, spike-in SILAC, spatial SILAC, and pulsed SILAC (pSILAC) to enhance its utility in quantification [159,160,161].

In neutron encoding or NeuCode SILAC, the mass defects of different stable isotopes within the same amino acid are used to encode multiple cell states and allow higher multiplexing [161,162]. In the super-SILAC method, a mixture of SILAC-labeled cells is combined. For instance, the combination of five SILAC-labeled cell lines with human carcinoma tissue generated hundreds of thousands of isotopically labeled peptides in appropriate amounts to serve as internal standards for MS-based analysis [159]. In spike-in SILAC, the non-labelled samples are combined separately with the SILAC standard followed by MS. A ratio of the sample relative to the standard is calculated. The differences between the samples are calculated by assessing their relative ratios. The spike-in SILAC has expanded to the quantitative analysis of tumor tissue samples in addition to cell culture [163,164,165]. The spatial SILAC is effective in tracking proteins to their original locations using distinct isotopic signatures that are introduced into discrete spatial cellular populations. The SILAC labels are individually pulsed to discrete positions, without altering the proteome [166]. In case of the pulsed SILAC [167], pulsed applications are used for temporal analysis. pSILAC monitors the initial incorporation of a heavy SILAC label in the surplus unlabeled medium over a period of time, which allows the assessment of the rate of protein label integration, and thus tracks the proteomic changes. These innovations have led to numerous SILAC-based temporal and spatial labeling applications [160].

SILAC is considered a precise quantitative method [168], as it allows the mixing of differentially labeled samples early in the experimental workflow that reduces variable sample losses at each step. It can be applied to intact proteins and could enable robust, multiplexed quantitation for top-down experiments [161]. The advancements in SILAC have enhanced our understanding of cancer biology and serve as a tool for biomarker discovery [169,170,171,172,173,174,175].

##### Isotope-Coded Affinity Tag

The isotope-coded affinity tag (iCAT) is an in vitro isotopic labeling method used for quantitative proteomics by MS [176,177]. Chemical labeling reagents are used to label and compare two samples. It consists of three elements: an affinity tag to isolate labeled proteins/peptides (e.g., biotin), a linker that incorporates stable isotopes (e.g., 9x C^13^ residues for heavy tag), and a reactive group for labeling an amino acid side chain (e.g., iodoacetamide to modify thiol group—cysteine residues) [178]. For the quantitative comparison of two proteomes, samples are separately labeled with the isotopically heavy (C^13^) and the isotopically light (C^12^) control [179,180]. The two different forms of the tag result in the mass difference between samples. Both samples are then combined, protease digested (e.g., trypsin), and subjected to affinity (e.g., avidin) chromatography to isolate peptides labeled with isotope-coded tagging reagents utilizing the affinity (e.g., biotin) tag, which are then analyzed by LC-MS to determine the *m/z* ratio between the proteins. The quantification of the ratios of signal intensities of differentially mass-tagged peptide pairs determines the relative levels of protein products in the two samples [178,179,180].

For locating cysteines involved in disulfide bonds, the tags are used to label all free cysteines prior to reduction/alkylation (iodoacetamide) of cysteines in disulfide bonds; as a result, the cysteine tag type indicates those involved in disulfide bonds. Although the iCAT greatly simplifies a complex tryptic digest by looking at only the cysteine-containing peptides, proteins with no cysteine cannot be quantitated. This becomes the main disadvantage of the iCAT labeling technique that it only analyzes cysteine-containing peptides, which constitute only a subset of the peptides (approximately 1% of the protein composition); as a result, the sites of PTMs and some proteoforms information might be lost. Moreover, iCAT becomes expensive if multiple samples are analyzed since only two labels are available. Despite these disadvantages, the iCAT coupled MS/MS is applied for both large-scale analysis of complex samples, such as whole proteomes, as well as small-scale analysis of subproteomes, and is widely utilized in oncoproteomics for protein identification and quantification [180,181,182,183].

##### Isobaric Tags for Relative and Absolute Quantification

Isobaric tags for relative and absolute quantification (iTRAQ) is a mass-tagging reagent that utilizes isobaric reagents to label the primary amino groups on the side chain of lysine residues and the N-terminus of tryptic peptides and proteins [184,185,186]. The iTRAQ reagents usually consist of three elements: a reporter group (an N-methyl piperazine), a balance group, and a reactive group with the primary amines of peptides (N-hydroxy succinimide ester). The balance group makes the labeled peptides from each sample isobaric and the analysis of the reporter group (generated by fragmentation in the mass spectrometer) enables the quantification. Currently, iTRAQ 4-plex (up to four different samples) and 8-plex (up to eight different samples) are used to label all peptides simultaneously in the samples. Typically, the proteins are extracted from different cells/treatment conditions and digested by a protease (e.g., trypsin) to generate proteolytic peptides. The peptide digests are labeled with different iTRAQ reagents to generate isobaric tag peptides. The labeled digests are pooled into one sample mixture and then subjected to identification and quantification by nano LC-MS/MS [186,187].

The iTRAQ-labeled peptides are isobaric and indistinguishable before peptide fragmentation (they produce only a single peak in a LC-MS scan). This is because each tag adds an identical mass to all peptides. MS/MS is used to generate fragmentation data, which can be searched in the available databases to identify the labeled peptides and hence the corresponding proteins. The reporter ions are specific for each of the different labels that are generated from fragmentation during MS/MS, resulting in the separation of different mass tags. The intensity ratio of the different reporter ions is used to relatively quantify the peptides and the proteins from which they originated. The disadvantages of iTRAQ are inconsistent labeling efficiencies, high cost of the reagents, and limited dynamic range for quantitative proteomics [184,185,186]. The use of standard operating protocols (SOPs) can produce reproducible and reliable results with iTRAQ by reducing the potential variability in multistep sample preparations. In spite of these shortcomings, the iTRAQ technique has high sensitivity and is ideally suited for normal/diseased/drug-treated samples comparison, time course studies, relative quantitation, PTM detection, biomarker discovery, and identification of proteoforms levels [74,188,189,190,191,192,193,194,195].

##### Tandem Mass Tag

Tandem mass tag (TMT) is a chemical labeling approach that uses isobaric mass tags, which are a set of molecules with the same mass that generate reporter ions of differing mass after fragmentation. The relative abundance of the tagged molecule can be determined by the relative ratio of the reporter ions. The TMT consists of four regions: a mass reporter region, a cleavable linker region, a mass normalizer region (spacer), and a protein reactive group (an amine-reactive NHS ester group). The chemical structures of all the tags are identical; however, each has isotopes substituted at various positions, such that the mass reporter and mass normalizer regions have different molecular masses in each tag. However, the total molecular weights and structure of the combined four regions of the tags remain the same. As a result, the molecules labeled with different tags are indistinguishable in chromatographic separation and single MS. However, the fragmentation of the tags in MS/MS gives rise to different mass reporter ions resulting in quantification [196,197].

The common varieties of TMT include TMT zero (a non-isotopically substituted core structure), TMT duplex (an isobaric pair of mass tags with a single isotopic substitution) [197], TMT 6-plex (an isobaric set of six mass tags with five isotopic substitutions) [198], and TMT 10-plex or 11-plex (a set of ten or eleven isotopic mass tags which use the TMT 6-plex reporter region that contain different numbers and combinations of 13C and 15N isotopes in the mass reporter) [199]. The recent TMT pro labels differ in structure, having different reporter regions (isobutyl proline mass reporter) and mass normalizer regions (longer spacer) than the original TMT [196]. The MS/MS fragmentation of the TMT pro tag produces a unique reporter mass of 126–134 Da in the low-mass region of the high-resolution MS/MS spectrum that facilitates the relative quantitation of proteoform abundance levels. The TMT pro labels include TMT pro zero, TMT pro 16-plex (a set of 16 isotopic mass tags), and TMT pro 18-plex (a set of 18 isotopic mass tags). The labeling efficiency, peptide/protein identification rates, and quantitative precision provided by TMTpro is the same as the original TMT reagents but provides improved quantitative accuracy with larger sample sets.

Usually, samples of equal abundance are labeled with TMTs. The isobarically labeled samples are referred to as isobaric carriers. The analytical sensitivity for all samples increases when one of the labeled samples is more abundant [200]. The publicly available unimodal database contains the structures of TMTs. The TMT facilitates sample multiplexing in MS-based quantification and identification of biological macromolecules, including proteins, peptides, and nucleic acids [201,202,203,204,205]. However, a common limitation in TMT is ion suppression due to coelution of TMT-labeled ions, resulting in simultaneous isolation and fragmentation of the interfering ion). In spite of the detrimental effect of the ion suppression on the accuracy, TMT-based quantification provides a higher precision than label-free quantification [202,203,204,206]. TMTs, in addition to protein quantification, increase the detection sensitivity of certain highly hydrophilic analytes, such as phosphopeptides [207,208]. TMTs are widely applied in oncoproteomics analysis [205,209,210,211,212,213].

##### Dimethyl Labeling

The stable isotope dimethyl labeling is a quantitation method that uses reductive amination [214]. In this strategy, combinations of several isotopic pairs of formaldehyde and cyanoborohydride are used to convert all primary amines (the N-terminus and the side chains of lysine, i.e., the epsilon-amino group of lysine residue) in proteins or peptides to dimethylamines [215]. This labeling produces peaks that differ by 28 Da for each derivatized site compared to its nonderivatized counterpart. By using a combination of isotopic pairs, proteins or peptides can be obtained that differ in mass by four Da between different samples. It has been successfully applied in the comparison of proteomes, phosphoproteomes, and affinity purification results [216,217]. Originally, this labeling strategy was used to conduct 2- or 3-plex quantitative proteomics analysis; however, later the use was extended up to 5-plex [218,219]. However, the mass difference between each of the derivatized forms becomes one Da for peptides with N-terminal proline and no internal lysine residues, and two Da for peptides with a single primary amine [219]. This results in substantial overlap from the natural abundance isotopes in the peptide.

This inexpensive, simple, and fast labeling strategy can be applied to a variety of samples such as tissue, lysate, or body fluids. Dimethyl labeling has been applied in various oncoproteomics studies [220,221].

##### Proteolytic ^18^O Labeling

In proteolytic ^18^O labeling, proteolytic catalysis is used to introduce two ^18^O atoms into the carboxyl termini of peptides in mixtures. Proteins are tryptic digested to generate peptide products that are dried and subsequently labeled [222,223]. For labeling, the peptides are redissolved in ^18^O-enriched water in the presence of trypsin [223]. In sufficiently enriched water (H_2_^18^O), the incorporation can exceed 95%. The catalytic enzyme can be immobilized on beads to facilitate its removal and terminate the exchange process, leaving water as a by-product. In differential (^16^O/^18^O) proteomics analysis, the samples are first separately labeled in H_2_^16^O and H_2_^18^O to produce labeled ^16^O- and ^18^O- peptides. In both samples, the heavy isotope-labeled (^18^O) peptides and the ^16^O-labeled peptides are then combined in a 1:1 ratio in presence of protease for differential ^16^O/^18^O coding followed by chromatographic and mass spectrometric analysis. The technique relies on the enzyme-catalyzed oxygen exchange and ^18^O exchange, where two ^16^O atoms are usually replaced by two ^18^O atoms in the presence of H_2_^18^O. The differentially labeled peptide ions exhibit a 2–4 Da mass shift, which can be measured by MS. This permits the identification, characterization, and relative quantitation of proteins from which the peptides are proteolytically generated [224]. It is used in comparative proteomics to quantitatively examine proteoforms abundance, PTMs, and to investigate interaction partners [225,226,227,228,229].

The ^18^O labeling is simple with limited sample manipulations and much cheaper than iCAT and SILAC, evaluating the price of reagents needed to label proteins. It is amenable for labeling samples with limited availability, such as human tissue specimens [230]. However, there are two disadvantages of ^18^O labeling. First, the inhomogeneous incorporation of ^18^O atoms into peptides results in a mixture of peptides having one ^16^O and ^18^O or both ^18^O oxygen atoms exchanged at their C-termini. The variable ^18^O incorporation alters the natural isotopic distribution and forms a complex isotope pattern, complicating the calculation of the ^18^O/^16^O ratios. The ^18^O incorporation is affected by various factors, including varying enzyme substrate specificity, oxygen back-exchange, pH dependency, and peptide physiochemical properties. The second disadvantage is the incapability to compare multiple samples within a single experiment. To circumvent the problem of variable ^18^O incorporation atoms, a true single ^18^O atom-labeling technique and a true two ^18^O atom-labeling technique needs to be developed [224,228,229].

Unlike iCAT, ^18^O labeling does not favor peptides containing certain amino acids (e.g., cysteine) and averts the requirement of additional affinity purification to enrich these peptides. Unlike iTRAQ, ^18^O labeling does not depend on fragmentation spectra (MS/MS) for quantitation. Thus, the inherent simplicity of this technique coupled with recent advances in the homogeneity of ^18^O incorporation and improvements in algorithms employed for assessing ^16^O/^18^O ratios makes it suitable for proteomic profiling of human specimens (e.g., plasma, serum, and tissues) in the field of biomarker discovery [230,231,232,233,234,235,236,237].

The selection of the isotope labeling technique is highly reliant upon the scope of analysis, the experimental design, and the sample/system being examined. These methods have the advantage of minimizing disparities between individually handled samples. However, the reagents are expensive, and the proteins may be partially labeled.

##### Label-Free

Label-free quantification determines the relative protein abundance among samples without any labeling procedures. The label-free approaches can be divided into distinct categories based on data extraction methods. The quantification can be either performed by (i) spectral counting, where the number of spectra assigned to a given peptide/protein are counted, or (ii) through the comparison of the peak intensity of the same peptide (MS1 signal intensity) or extraction of the area of the precursor ions’ chromatographic peaks, called the area under the curve (AUC) [238,239,240].

In spectral counting methods, the relative protein quantification is performed by measuring the frequency with which the protein/peptide of interest is identified by the MS spectra, which may directly correlate with the protein product abundance. An increase in protein product abundance results in an increase in the number of spectra for its proteolytic peptides. The increase in the number of digests results in an increase in protein sequence coverage, the number of identified unique peptides, and the number of identified total spectral counts (MS spectra) for each protein product. For accurate and reliable detection of protein products in complex mixtures, normalization and statistical analysis of spectral counting datasets are performed. In an LC-MS/MS experiment, the exponentially modified protein abundance index (emPAI) is used to estimate absolute protein abundances from peptide counts. The protein abundance index (PAI) is the ratio of observed peptides to the number of observable peptides per protein. The PAI is approximately proportional to the logarithm of absolute protein concentration [241,242].

In ion intensity methods, the signal intensity from the MS is correlated with ion concentration. The height or area of a peak at a particular *m/z* ratio from a mass spectrum reflects the number of ions (ion abundance) for that *m/z* detected by the mass spectrometer at a particular time. However, the ion abundance can only be used for relative quantification instead of absolute quantification since the ionization efficiency for each peptide is different. The differential expression can be calculated by the ratio of ion abundances between identical peptides in different experiments [240].

Label-free quantitation is easy to use, yields highly reproducible results, and is reliable [243,244,245]. It is cost-effective (avoids expensive chemical and metabolic tags) and allows the profiling of a number of large samples with the flexibility of multiple comparisons [246,247]. It eliminates the chance of variability that chemical labeling/tagging introduces and significantly reduces the sample preparation time by eliminating numerous steps [248]. It has an excellent linear dynamic range of about three orders of magnitude. Typically, the following steps are involved in label-free quantitative proteomics: sample preparation (protein extraction, reduction, alkylation, and digestion), separation by LC and analysis by MS/MS, and data analysis (peptide/protein identification, quantification, and statistical analysis). Based on the requirement, each sample is subjected to individual LC-MS/MS or LC/LC-MS/MS runs. The parallel sample handling results in a uniform treatment of the sample, which correctly attributes to the actual proteoform abundance differences between samples [238]. However, the measurement of small changes in the quantity of low-abundance proteoforms becomes difficult and often gets masked by sampling error, posing limitations in the analysis of changes in proteoform abundances in complex biological samples [249]. Additional concerns associated with label-free quantitation are sequence coverage and the extent of complex sample fractionations prior to MS analysis. The sample processing also requires normalization as run-to-run analysis of the samples can exhibit differences in the peak intensities of the peptides [239]. Further, experimental drifts in retention time and *m*/*z* may complicate the accurate comparison of multiple LC-MS data sets, multiple sample injections onto the same reversed-phase HPLC column may result in chromatographic shifts, and unaligned peak comparison may result in large variability and inaccuracy in quantitation. To solve these issues and automatically analyze the data at a comprehensive scale, various software packages have been developed [250,251]. These include the public domain software suites such as MaxQuant [252], Trans proteomic pipeline [253], and Skyline [240], or commercial software such as PEAKS [254], ProteinLynx (Water corporations), and Proteome Discoverer (Thermo Fisher Scientific).

The label-free quantitative method has been applied in proteomic profiling of different biological processes, diagnosing cancer biomarkers, and studying proteoform interaction networks [255,256,257,258,259].

### 2.3. Microarrays

Microarrays, also known as biochips, are a collection of microscopic biomolecules spotted on a solid support that are used to identify interacting partners via affinity interaction.

#### 2.3.1. Protein Microarray

A protein microarray (or protein chip) is a high-throughput tool for studying the biochemical activities of proteins, their interactions, and function determination on a large scale using miniaturized assays [260]. The main advantage is that large numbers of proteins can be followed in parallel. Typically, the chip contains numerous spots of either proteins or their ligands arranged in a predefined pattern, arrayed by robots onto a solid support surface. The support surface can be a glass slide, nitrocellulose membrane, bead, or microtiter plate, to which an array of capture proteins is bound [261]. Usually, fluorescent dye-labeled probe molecules are added to the array after sample application. Any reaction between the probe and the immobilized proteins emits a fluorescent signal that is measured by a laser scanner. Protein microarrays are quick, automated, cost-effective (require minuscule samples and reagents), and highly sensitive. Additionally, thousands of known proteins can be analyzed in a single experiment. Protein microarrays have become an indispensable tool for proteomic applications and multi-parameter clinical diagnostic tests [262]. Protein microarrays can be created in two formats: a forward phase or reverse phase.

In a forward phase protein array (FPPA), the different capture molecules are first immobilized on a solid surface to capture the corresponding bait molecule in a test sample (such as serum or cell lysate). The captured analyte is then detected directly with a fluorescent labeled detection probe or detected indirectly with the detection probe followed by a fluorescent labeled second probe. The disadvantages of FPPA include the requirement for two distinct probes directed against the same bait, time-consuming identification of a capture and a detection affinity probe, and the inability to match the probe affinities to the sample protein concentration.

In a reverse-phase protein array (RPPA), the bait molecule is directly immobilized on a solid support and detected with a single affinity probe either by colorimetric amplification or fluorescence detection. The bait molecule can be a protein present in a cell lysate, serum, or subcellular fraction [263]. By immobilizing the bait molecule in a dilution series, it is possible to effectively match the sample protein concentration with the probe’s affinity, allowing measurement within the linear dynamic range of the array. Large sample profiling can be performed in parallel to allow hundreds of targets to be interrogated in one experimental run [264]. Further, the minimal pre-experimental process increases sensitivity and permits subtle fluctuations to be monitored. RPPA assays are particularly suitable for identifying proteins, proteoforms, and PTMs including phosphorylation, methylation, and acetylation within signaling networks [265,266]. The disadvantage of RPPA is that the specificity might be compromised to some degree, as a single detection probe/antibody is used. The sophisticated workflow of RPPA requires array printing, multiple steps of immunostaining and signal amplification, high-resolution data capture, and data processing and analysis [263]. Multiplex discovery proteomics may further slow the turnaround time. Another difficulty is in the validation of RPPA-usable antibodies/probes due to the antigen-down immunoreaction format. In spite of the challenges, the minimal inter-assay variation makes RPPA suitable for cancer biomarker validation [267,268,269,270] and is used for large-scale patient profiling and diagnosis in various cancers [59,271].

#### 2.3.2. Antibody/Antigen Microarrays

In antibody microarrays, the specific capture antibodies are immobilized on a modified planar solid surface such as a nitrocellulose membrane, glass slide, silicone, or bead via covalent binding, affinity binding, or physical entrapment. The sample (such as serum or cell lysate) is then applied to detect the interaction between the antibody and its target protein. Antibody arrays, such as bead-based arrays and sandwich ELISA-based planar arrays, provide medium-/low-plex proteomic profiling. For high-plex profiling, samples are labeled with fluorescent, chemiluminescent, or oligo-coupled tags to allow differential signal amplification and detection. This method can practically characterize over a thousand proteins with minimal immunogenic cross-reactivity induced by antibodies [59].

Antibody arrays have very high performance for knowledge-based examinations, providing a high-throughput, semi-quantitative, or quantitative analysis. In contrast to untargeted proteomic approaches, it is highly sensitive. Ultramicroarrays have been developed to combine the advantages of multiplexing capabilities, higher throughput, and cost savings, with the ability to screen minuscule samples [272]. Antibody arrays are particularly useful for proteomic profiling of low-abundance proteoforms. It has been extensively applied in the high-throughput multiplexed analysis of cancer biomarkers [273,274].

In the antigen microarrays/functional protein arrays, ectopically expressed proteins/peptides with a wide range of proteome coverage in species of interest are arrayed on the support surface. These serve as baits to capture analytes of interest within the applied sample [275]. They can be used to investigate the interaction with proteoforms, lipids, small molecules, nucleic acids, and antibodies. For instance, serological autoantibodies (AAbs) for cancer biomarker profiling have been identified using high-plex protein arrays in ovarian, gastric, bladder, prostate, and breast cancers [275,276].

The disadvantages of antibody/antigen microarrays include that they are not discovery-oriented approaches, have narrow dynamic ranges, are restricted to the detection of usable and compatible proteins, have sample labeling prerequisites, their cost, shelf-life, and inter-assay variability [277]. In addition to the above limitations, finding a high-quality and specific antibody against every protein and its proteoform (e.g., phosphorylated and glycosylated) in the proteome is challenging. The platforms for high-throughput expression and purification with PTMs are necessitated for reproducible spotting of the complete proteome. Therefore, standard criteria for array production, data normalization, variance estimation, and analysis of proteoform abundance levels would improve the interpretation of microarray results. The difficulties in spotting protein into arrays led to the development of the nucleic acid programmable protein array (NAPPA). In the NAPPA, the cell-free extracts are used to directly transcribe and translate cDNAs encoding target proteins onto the solid support, such as glass slides. The advantage of the NAPPA is that it eliminates the need for protein purification, avoids storage-associated protein stability issues, and captures sufficient protein for functional studies [278,279]. The NAPPA coupled with MS has been used to identify peptide sequences for potential phosphorylation and to investigate protein–protein interaction [280].

Another advancement in protein arrays is the development of suspension bead arrays. The suspension platforms allow the identification of protein–ligand interactions in solutions. Suspension bead arrays are flexible to capture any protein–ligand interaction by coupling the required proteins or ligands to distinct bead populations. For instance, the Luminex beads enable simultaneous quantitation of up to a hundred different biomolecules in a single microwell plate. The suspension platforms, such as the LiquiChip system (Qiagen) or the Bio-Plex system (Bio-Rad Laboratories), use Luminex’s bead-based xMAP technology [281]. In Bio-Plex systems, differentially detectable bead sets are used as a substrate to capture analytes in solution and fluorescent methods for detection [282].

#### 2.3.3. Tissue Microarrays

The tissue microarray (TMA) is a high-throughput technology that enables simultaneous proteome analysis from thousands of individual tissue samples in a single microscopic slide [283]. It was first described by Kononen in 1998 [284]. The tissues are formalin-fixed and paraffin-embedded from which small cylindrical tissue cores as small as 0.6 mm in diameter from regions of interest that are extracted using hollow needles of set diameters and transferred into a matrix slot within a recipient paraffin block. Sections from each microarray array block are cut using a microtome into 50–1000 sections that can be subjected to independent tests on a microscope slide and analyzed by a variety of assay and staining techniques, including immunohistochemistry (IHC) and fluorescent in situ hybridization (FISH) analysis, in situ PCR, and cDNA hybridization. This facilitates the rapid analysis of hundreds of patient samples [285].

TMAs are useful for oncoproteomics studies, the development of diagnostic tests, the discovery of cancer biomarkers, laboratory quality assurance [286], and the assessment of histology-based laboratory tests (e.g., IHC and FISH) [286]. TMAs for discovery and nonclinical work are less stringently classified. Usually, formalin-fixed paraffin-embedded or frozen TMAs are used. In nonclinical settings, TMAs are helpful in assessing target protein distribution in a variety of tissues, which provides guidance on tissue selection to investigate the efficacy and toxicity studies for evaluating therapeutic effects. With specific antibodies, a comprehensive protein analysis can be performed. In the absence of suitable antibodies for use in formalin-fixed tissues, frozen TMAs are useful for IHC. Clinical TMAs are subclassified into (1) prognosis microarrays (samples from clinical follow-up data), (2) progression microarrays (samples of different stages of tumor progression within a given organ), (3) multi-tumor microarrays (samples from multiple histological tumor types, and (4) cryomicroarrays (frozen samples).

The TMAs are rapid, high-throughput, and have automated data reads. However, they involve laborious build-up, as the heterogeneous tumor tissues may require multiple punches to ensure ample representations of the sample analyzed. TMAs have been widely adopted in oncoproteomics analysis to identify novel biomarkers. For instance, Tenascin-C is identified as a novel candidate marker for cancer stem cells in glioblastoma [287]. TMAs are also used to investigate associations between the expression of specific tumor receptors and their tissue alterations in various cancers, such as breast cancer [288], bladder cancer [289], soft-tissue sarcoma [290], and prostatic cancer [291].

#### 2.3.4. Protein Domain Microarray

Protein microarrays are efficient in high-throughput identification and quantification of protein–protein interactions. However, proteins exhibit a wide range of physicochemical properties and often recombinant production is difficult. To sidestep these issues and to read the PTM signal placed on the interacting partners, families of protein interaction domains can be focused. Protein domains bind to short peptide motifs in their corresponding ligands to mediate protein–protein interactions. These peptide recognition elements are important for multiprotein complex assemblies. The protein domain microarray consists of protein interaction domains arrayed onto solid support, such as nitrocellulose-coated glass slides, to generate a protein–domain chip [292]. The arrayed domains retain their binding integrity for their respective peptides/protein. The high-throughput quantification of domain–peptide interactions can be performed using fluorescently labeled synthetic peptides [293]. For instance, protein domain microarrays of human Src homology 2 (SH2), Src homology 3 (SH3), phosphotyrosine binding (PTB) domain, a domain with two conserved tryptophans (WW), forkhead-associated (FHA), PDZ domains (a domain originally identified in PSD-95, DLG, and ZO-1 proteins), pleckstrin homology (PH), and a domain with two conserved phenylalanines (FF) domains have been produced [292,293]. In the case of domains that mediate high-affinity interactions, saturation binding curves can be used to measure equilibrium dissociation constants for their peptide ligands directly on arrays. For weaker binding domains, arrays can be used to identify candidate interactions that can be quantified by fluorescence polarization.

The protein–domain chip can also be used to identify interacting protein partners in a total cell lysate. These domain-bound proteins can then be detected using a specific antibody, generating an interactive map for a protein of interest. The protein–domain chips can identify qualitative differences in protein ligands caused by PTMs and rapidly quantify protein–ligand interactions, even with minuscule samples. The simultaneous cross-examination of entire domain families provides a potent way to evaluate binding selectivity on a proteome-wide scale and unbiased information on the connectivity of protein–protein interaction networks [294].

#### 2.3.5. Immunosensor Arrays

Immunosensor arrays are a type of affinity-based biosensors that detect a specific target analyte or antigen by the formation of a stable immunocomplex between the antigen and the capture antibody, which results in the generation of a measurable signal by a transducer. The use of antibodies as molecular recognition agents provides ultrahigh specificity in immunoassay and facilitates the detection of cancer biomarkers [295]. For cancer diagnostics, the immunoassay is integrated with several detection strategies, such as fluorescence [296], colorimetric [297], plasmon resonance sensors [298], electrical [299], optical [299], electrochemical [300], chemiluminescence [301], and electrochemiluminescence [302].

One aspect of oncoproteomics is directed toward the development of accessible and ultra-sensitive cancer diagnostic tools that rely on protein biomarkers associated with various cancer that are overexpressed in body fluids. Protein biomarker detection for point-of-care use requires highly sensitive, non-invasive microfluidic cancer diagnostics that can overcome the limitation of low sensitivities imposed by imaging and invasive biopsies. Electrochemical immunoassays have become popular as protein detection methods due to their inherent high sensitivity and ease of coupling with 3D printed electrodes. Integrated chips with printed electrodes can be built at a low cost and designed for automation. Three-dimensional printing also known as additive manufacturing is being utilized to develop user-friendly, semi-automated, and highly sensitive protein biomarker sensors at low-cost. These can be tailored toward clinical needs [303]. Most of these ultrasensitive detection systems use enzyme-linked immunosorbent assay (ELISA) features with microfluidics that permits easy manipulation and good fluid dynamics to deliver reagents and detect the desired proteins [304]. The fabrication, as well as validation of a novel 3D-printed, low-cost, automated miniature immunoarray has been reported that detects multiple proteins with ultralow detection limits [305]. It uses electrochemiluminescent detection with a CCD camera. The automation is facilitated by a touch-screen control of the micropump. The prefilled reservoirs deliver the sample and reagents to a paper-thin pyrolytic graphite microwell detection chip to complete sandwich immunoassays. The high sensitivity of the detection chip is achieved via the use of single-wall carbon nanotube antibody conjugates in the microwells and enormously labeled antibody decked RuBPY silica nanoparticles to generate electrochemiluminescence. It can detect eight proteins of a prostate cancer biomarker panel in human serum samples in 25 min [305]. The microfluidic platform has also been used for the generation of cancer spheroids on a chip (large arrays of breast tumor spheroids), grown under close-to-physiological flow in a biomimetic [306]. The on-chip spheroid drug response can be correlated with the in vivo drug efficacy. Thus, it can be used for time-, labor-, and cost-effective investigations of the effects of drug dose and supply rate on the chemosensitivity of cancer cells. Overall, the multiplex immunosensor and microfluidic arrays have entered clinical, point-of-care diagnostic testing, and resource-limited environments [307,308].

## 3. Contemporary Technologies and Approaches

The following section describes emerging proteomics technologies that could play an important role in cancer diagnosis and treatment.

### 3.1. Laser Capture Microdissection

Laser capture microdissection (LCM) is an effective extraction technique to harvest pure subpopulations of cells from tissue sections under direct visualization of a microscope. The cells of interest are harvested either directly or by cutting away unwanted cells to obtain histologically pure enriched specific cell populations. LCM has expanded the analytical capabilities of proteomics to analyze proteins from extremely small samples [309,310]. It basically allows for the miniaturization of extraction, isolation, and detection of hundreds of proteins from different cell populations containing only a few cells. However, as the sample size decreases, each step requires care. The LCM dissected tissues are subjected to protein extraction, reduction, alkylation, and digestion, followed by injection into a nano-LC MS/MS system to simultaneously identify and quantify hundreds of proteins. The validation can be performed by secondary screening using immunological techniques, such as IHC or immunoblots [309,311]. The advancement in LCM technology enables effective high-throughput sampling of specific cellular subtypes [311]. LCM-coupled 2D-DIGE and/or quantitative MS approaches have been used for proteomics analysis of distinct, pure cell populations [312,313,314] and to investigate various cancer-associated protein profiles [311,313,314,315,316].

### 3.2. Aptamer-Based Molecular Probes for Protein Signature of Cancer Cells

Aptamers are a class of short, single-stranded DNA, RNA, or peptide (~25–80 nucleotides/amino acids) that after acquiring a specific tertiary structure, bind to various targets with high affinity and selectivity. Aptamers are also known as a ‘chemical antibody’ and possess several intrinsic advantages, such as convenient modification, easy synthesis, good compatibility, and high programmability. A process known as systematic evolution of ligands by exponential enrichment (SELEX) was first used to screen aptamers. Aptamers can be generated against various targets, such as small molecules, peptides, proteins, and intact living cancer cells. Some of the examples of aptamers used against cancer cells are Sgc8, specific for acute lymphoblastic leukemia [317] and XQ-2D, for pancreatic ductal adenocarcinoma [318].

In a recent study to identify CRC patients from healthy controls or adenoma, 1317 protein-based aptamer screenings were used in a liquid biopsy [319]. Another aptamer-based study measured 813 proteins in 1326 serum samples of non-small cell lung cancer (NSCLC) and healthy individuals that identified multiple potential biomarkers for early detection of NSCLC [320]. This study led to the development of a 7-protein biomarker panel in clinical settings (AptoDetect-Lung) [321]. Currently, more than 7000 protein-specific aptamers are being used for commercial assays. A limitation of this technology is to develop high-quality aptamers for novel targets. In addition, aptamers used to detect the post-translational modification are still in the preliminary stage; although, some phosphor-specific aptamers have been developed.

### 3.3. Extracellular Vesicle-Based Protein Blood Test

Tumor-derived Extracellular Vesicles (EVs) have recently emerged as an important biomarker in blood circulation for the diagnosis of cancer. The EVs are nano-/micro-meter size lipid bilayer-enclosed vesicles that contain various molecules, such as proteins, nucleic acid, and lipids from parental tumor cells. Tumor-derived EVs are present in abundance in blood circulation compared to other biomarkers, as they release in blood 10^4^ quantities per day. The proteins enveloped in EVs play an important role in cancer metastasis and progression, including immune evasion, matrix remodeling, tumor vascularization, and premetastatic niche formation. Detection of EV protein markers facilitate the diagnosis and monitoring of various cancers.

EV isolation and analysis are considered to be difficult because of their very small size and low densities. To improve the recovery (yield) and specificity (purity), many commercial kits and new techniques have been developed for the isolation of EVs from various biological specimens. The population of EVs is heterogenous in the biological specimen, which further varies with the size distribution and physicochemical properties such as solubilities, surface identities, charge, and hydrodynamic properties. All these properties of EVs are taken into consideration during the isolation and purification process. The common isolation and purification methods involved are ultracentrifugation, precipitation, ultrafiltration, size exclusion chromatography, affinity interaction, microfluidic devices, and microchips.

Melo et al. showed the presence of proteoglycan glypican-1 (GPC-1) positive EVs in the serum of pancreatic cancer patients, which correlated the level of GPC-1 with the tumor burden and survival of pre- and post-surgical patients, suggesting it as a prognostic marker [322]. Lai et al. concluded EV GPC-1 as a diagnostic marker for pancreatic cancer using the LC-MS/MS method [323]. Further, EV Del-1 was detected as a promising marker for breast cancer [324]. Three studies using EV arrays identified a 30-marker model and a 10-marker model for the diagnosis of NSCLC and adenocarcinoma, respectively [325,326,327]. Another report showed that by using a serum-based amplified luminescent proximity homogeneous assay for EV detection, CD147 and CD9 double-positive EVs were significantly higher in the serum of colorectal cancer patients, as compared to healthy donors [328]. From all the above examples, we can say that an extracellular vesicle-based protein blood test provides highly promising biomarkers candidates for the diagnosis and prognosis of cancers.

### 3.4. Proximity Extension Assay

The proximity extension assay (PEA) is a combination of two sandwich ELISAs and highly specific and sensitive polymerase chain reaction (PCR) technologies that detect protein–protein interaction and liquid biopsy-based discovery in cancer. It has a broad dynamic range of 10 logs and minimal sample requirement, which makes it a very useful tool for serological profiling. In PEA, multiple antibodies are pooled with the protein of interest. Each antibody in a pair is attached with complementary DNA oligonucleotides that allow hybridization when the correct antibody pairs come close together by binding to the target protein. The resultant double-stranded DNA is PCR amplified and is used to measure the relative concentration of the target proteins. The most recent commercial PEA assay has standard measurement coverage of 3072 target proteins.

PEA was first used to identify the prognostic biomarkers from the blood in colorectal cancer [329]. It was also extended for other cancer types for blood proteomic profiling such as cervical, ovarian, prostate, lung, and hematopoietic cancer for early detection, diagnostics, and disease monitoring [330,331,332,333,334]. The PEA has a wide dynamic range with high accuracy and reproducibility within the pg/mL range. The applications of the PEA in liquid biopsies, including cellular lysates, open new avenues for integrative multi-omics profiling.

### 3.5. Immuno-Affinity Capillary Electrophoresis

Immuno-affinity capillary electrophoresis (IACE) is an emerging powerful diagnostic tool to isolate, separate, detect, and characterize proteoform in biological fluids. It combines the power of highly selective capture agents with the high resolving power of capillary electrophoresis. IACE separates the substances by specific and non-specific noncovalent affinity interactions during electrophoresis. The interacting target molecule is captured and bound to affinity reagents onto the wall of capillary or solid support. Then, the remaining sample is removed, and the target molecule is released using an elution buffer. As a result, the target molecule becomes purified and concentrated significantly in the solution. IACE uses antibodies as affinity reagents, however other variants have emerged that use various ligands/probes (e.g., lectins, aptamers, and metalorganics), for selective capturing of the target. IACE coupled with MS has been used for mapping proteoform in various studies [335,336,337]. It is used as a point-of-care instrument to profile proteoform patterns in biological fluids, and in proteins obtained from tumor cells, exosomes, or vesicles present in biofluids [336,337,338,339,340,341].

### 3.6. Cancer Immunomics to Identify Autoantibody Signatures

Antibodies associated with cancer develop early during carcinogenesis when cancer-associated antigens appear in premalignant or malignant tissue. The cancer antigens are recognized by the effective immune response of autoantibodies, which makes autoantibodies a suitable biomarker for cancer detection. For example, autoantibodies against HCC1, CDKN2A, P53, the cellular inhibitor of PP2A (CIP2A), and the cyclin-dependent kinase inhibitor 2A (CDKN2A) indicate the presence of HCC prior to its clinical diagnosis [342]. These autoantibodies can be detected by proteome analysis in serum by using 2DGE, immunoblotting, image analysis, and MS [342]. Alternatively, a new approach has been developed to detect the autoantibodies in cancer patients by combining the 2D immunoaffinity chromatography, enzymatic digestion of the isolated antigens, nanoflow separation of the resulting peptides, and identification: MAPPing (multiple affinity protein profiling) [343].

### 3.7. Protein Terminomics

Proteases are key enzymes involved in protein terminomics. Proteases regulate vital biological processes of apoptosis, neurodegeneration, infection, and cell differentiation. Proteolysis performed by proteases is an important post-translational modification of a protein. Around 600 human proteases are reported and categorized into five families based on their catalytic mechanisms (threonine, serine, cysteine, metallo, and aspartyl proteases). There are two methods commonly used in protein terminomics: N-terminomics and C-terminomics. N-terminomics involves the labeling of the proteolytic protein fragments and the enrichment of the fragments from the complex mixture. The enrichment can be achieved by adding the functional group or by labeling with the isotope to the cleaved peptide. As C-terminal labeling is quite difficult as compared with N-terminal labeling, the N-terminomics is widely used. In N-terminomics, some methods are established, such as the COFRADIC (combinatorial fractional diagonal chromatography), subtilligase, and TAILS (terminal amine isotopic labeling of substrates) methods [344]. N-terminomics has been used to identify the substrate of neutrophil-specific membrane-type 6 matrix metalloproteinase (MTP6-MMP), which plays a role in cancer [345]. Moreover, Alcaraz et al. used TAILS to identify the substrate of cathepsin D, which is a tumor-specific protease in triple-negative breast cancer cells [346].

### 3.8. Single-Cell Proteomics

Cancer tissue shows multiple genomic variations and heterogeneity at the level of proteome. This cell-to-cell variability is responsible for the altered biomarker expression in the different cells of the same tissue that may be overlooked when biomarker quantitation is based on the bulk tissue sample. Single-cell proteomics allows measuring of the level of prognostic and diagnostic biomarkers at the level of a single cell of a cancerous tissue that provides information about a single-cell subpopulation carrying cancerous characteristics. This information can further be used for patient risk stratification and individualized therapy. The techniques which are being used for single-cell proteomics are as follows: microfluidics and laboratory-on-a-chip technology, flow cytometry, mass cytometry, and chemical cytometry.

To investigate the different combinations of drugs in different cellular subsets during the course of cancer therapeutic treatment, several ex vivo screens have been developed where the bone marrow cells from AML patients were challenged with an array of therapeutic drugs [347,348]. To evaluate the efficacy of a large number of clinically approved drugs in AML and healthy cells, a drug sensitivity and resistance testing (DSRT) assay was developed that identified the drugs which selectively target the leukemic cells [348]. Similarly, Bennet et al. designed an automated assay where bone marrow samples from AML patients were treated with clinically approved drugs and cell identity was determined by flow cytometry [347].

### 3.9. Nanoproteomics

The complexity of the proteome challenges and the new methods to detect the small number of proteoforms present differing concentrations. To detect the low abundance proteoforms that can be isolated from the limited source material (e.g., biopsies), the nanoproteomics platform provides improved specificity, reproducibility, biocompatibility, and robustness compared to the current conventional proteomic techniques. Nanoproteomics can be defined as the application of nanobiotechnology to proteomics. It uses nanoscale devices such as nanofluidics and nanoarrays. Unique nanomaterials such as quantum dots (QDs), carbon nanotubes (CNTs), and gold nanoparticles (GNPs) are being used in nanoproteomics techniques. The application of nanoproteomics techniques in cancer advances the discovery of biomarkers and detection of early cancer pathogenesis. Unger et al. used NanoPro 1000, a rapid and highly sensitive immunoassay platform, to show the phoshphorylation status of clinically relevant cancer-related biomarkers in response to ischemia in tissue samples from primary colorectal cancer patients [349].

### 3.10. PTM Enrichment Methods

Post-translation modifications (PTMs) comprise phosphorylation, acetylation, methylation, glycosylation, ubiquitination, and SUMOylation (among other modifications), and because of their low abundance and labile nature, enrichment of a modified protein product is required for MS analysis.

The enrichment of proteoforms can be achieved by affinity or chemical strategies prior to MS. Affinity strategies require antibody/protein domain recognition for purification or chromatographic separation based on specific properties of the PTM, while chemical enrichment strategies involve chemo-selective probes, metabolic labeling by unnatural precursors, and chemoenzymatic labeling. One example of chemical enrichment is the use of streptavidin beads to interact with the biotin-tagged protein. Phosphopeptide enrichment by affinity approaches depends on the interaction of phosphorylated amino acid with different binding reagents and is categorized into ion exchange chromatography [350], affinity chromatography [351], and antibody/protein domain-based enrichment of phosphor-tyrosines [351]. For acetylated protein enrichment, immunoaffinity purification using pan-anti-acetyl-lysine antibodies is widely used [352]. A combination of different enrichment or fractionation strategies improves the enrichment. For example, immune enrichment with OFFGEL isoelectric focusing (IEF) separation or a combination of strong cation exchanger (SCX) with immuno-precipitation or COFRADIC [352]. For enriching methylation *PTMs,* various pan-methyl antibodies that recognize mono and symmetric R dimethylations have been developed and commercialized [353]. For glycosylated proteoform enrichments, affinity strategies use the different sugar recognition specificities of carbohydrate-binding proteins, such as lectin. The most widely used lectins in glycoprotein enrichment are *Sambucus nigra*, concanavalin A, wheat germ agglutinin, and *Ricinus communis* agglutinin. In chemical enrichment methods for glycoproteins, labeling with azido and alkynyl monosaccharide precursors, such as variants of GlcNAc, GalNAc, N-acetylmannosamine (ManNAc), and fucose are used [354]. For ubiquitinated and sumoylated proteoform enrichment, affinity-based strategies are available to date. Different tags, such as His-, hemagglutinin-, or biotinylated-Ub, are used for the enrichment of ubiquitinated proteoforms [355].

## 4. Role of Proteomics in the Prognosis and Diagnosis of Cancer

Proteomics investigations can be divided into two major areas: expression proteomics and functional proteomics. Expression proteomics deals with the up and downregulation of protein levels. Functional proteomics defines the molecular mechanism and discovers the unknown function of a protein [356,357]. It includes PTMs, characterization of protein complexes, and enzyme activities. The detection of various proteoforms by functional proteomics helps to identify therapeutic targets and diagnostic biomarkers in cancer.

HeLa cells are the most commonly analyzed immortalized cancer cell line that has been used for decades in proteomics studies. It is a cervical adenocarcinoma cell line that expresses over 10,000 proteins. The tryptic digest mixture of HeLa S3 cells is commercially available. It has been used as the most widely characterized MS standard in proteomics. HeLa S3 cells contain various PTMs that have been used for the method development of phosphoprotein analysis, stable isotope-coded expression proteomics studies, and glycopeptide enrichment [358]. There are studies available in the literature that discuss diverse aspects of biomarker discovery [359,360,361,362,363,364]. In the coming sections, we have discussed the application of proteomic technologies in the discovery of prognostic and diagnostic biomarkers in various cancers (Figure 2).

### 4.1. Hepatocellular Carcinoma

Hepatocellular carcinoma (HCC) is the most common form of primary malignant tumor of the liver that increases morbidity and mortality [365]. Among cancers, HCC is diagnosed fifth in the world, while it ranks as the second and sixth most frequent cause of cancer-related death in men and women, respectively [366]. In the early stage of HCC, it is accompanied by the hepatitis B virus (HBV) and the hepatitis C virus (HCV), which is followed by liver cirrhosis, the main cause of HCC [367,368]. The higher death rate in HCC is attributed to the lack of reliable diagnostic and prognostic markers and limited treatment options. Therefore, more specific and sensitive biomarkers are required to evaluate the disease progression and metastasis risk to predict cancer recurrence [369].

HCC is diagnosed by liver biopsy analysis or by cross-sectional imaging techniques, such as contrast-enhanced computer tomography (CT) and magnetic resonance imaging (MRI) [370,371,372]. However, these imaging techniques are more time-consuming and less sensitive, which directs to the development of novel screening methods to detect specific biomarkers of HCC with higher sensitivity. Here, we have summarized (Table 1) the recent methodological development in proteomics approaches to detect the diagnostic and prognostic biomarkers of HCC.

### 4.2. Colorectal Cancer

Colorectal cancer (CRC) is an aggressive form of tumor. It is the second leading cause of cancer death (9.4%) and is the third most common cancer (10%) worldwide [384]. CRC prognosis is very poor and around 60–65% survive within five years after diagnosis [385], which further drops if metastasis occurs [386,387]. The occurrence of primary tumor metastasis is responsible for 90% of CRC deaths. The liver is the most common site of CRC metastasis, which is also referred to as colorectal liver metastasis (CRLM). This is because of the portal venous drainage from the colon and rectum to the liver [388,389].

Currently, pathological staging of tumors is considered a gold standard for CRC prognosis [390]; however, it fails to predict the recurrence in patients undergoing surgical resection for colorectal cancer treatment [390]. Given that some molecular mechanisms controlling colorectal carcinogenesis and its metastasis have been identified, there is a need to develop novel diagnostic and prognostic tools along with new therapies for colorectal cancer diagnosis and treatment. Proteomics-based techniques have emerged as a promising approach for the identification of prognostic and diagnostic biomarkers for CRC. We have summarized below (Table 2) the recent methodological development in proteomics approaches to detect the diagnostic and prognostic biomarkers of CRC.

### 4.3. Leukemia

Acute myeloid leukemia (AML) is an aggressive form of leukemia that is heterogenous in nature [411,412]. Around half of the patients who are diagnosed with AML and achieved complete remission after intensive and potentially curative treatment relapse within the next three years, constituting a 50% survival rate [413,414]. Relapsed AML patients are treated with salvage cytotoxic therapy along with currently available clinical tests, such as pathway-targeted agents and immunotherapy-based approaches [415] that have limited efficacy. Thus, there is a need for better prognostic and therapeutic strategies for the large majority of leukemia patients.

Recently, various proteomics approaches have played an important role in the diagnosis or prognosis of leukemia. Phosphoproteomics or LC-MS/MS-based proteomics has been used for the staging of patients with AML [416,417,418]. Advancements in the MS-based approaches have provided the optimized resolution for the high coverage and characterization of PTMs, and the description of the tyrosin kinome, tyrosine phophatome, and phosphotyrosine proteome is a predictive phosphorylation marker [419]. We have provided below (Table 3) various studies that involve different proteomics approaches to identify the diagnostic and prognostic markers of AML.

### 4.4. Prostate Cancer

Prostate cancer (PCa) is the fifth cause of death worldwide and the second most commonly occurring cancer in men [432]. Among various causes of prostate cancer androgen receptor (AR) signaling, PTEN/PI3K/AKT/mTOR pathway, IL6/STAT3 and STAT5a/b are the most common pathways involved in prostate cancer cell survival and resistance [433,434]. Because of the involvement of androgens and AR, androgen deprivation therapy and downregulation of AR signaling are the most common therapeutic approaches [435]. The current diagnostic methods involve transrectal ultrasound, prostate-specific antigen (PSA) blood levels, digital rectal examinations, and prostate biopsies. However, these methods are invasive, expensive, and frequently lead to false positive or false negative results [436].

Modern proteomics technologies have emerged as a new detection, management, and surveillance tool for the discovery of new biomarkers of prostate cancer. Because of the only available biomarker PSA for prostate cancer, there is an urgent need to discover new biomarkers that will lead to personalized and targeted therapies. The following table summarizes (Table 4) various proteomics approaches used to identify the prognostic and diagnostic markers of prostate cancer.

### 4.5. Lung Cancer

Lung cancer is the most prevalent form of cancer in the world and is reported as one of the main cause of mortality [462]. The most common type of lung cancer is small-cell lung cancer (SCLC) and non-small cell lung cancer (NSCLC) [463]. About 13% of lung cancers are SCLC and 84% are NSCLC. The five-year relative survival rate for NSCLC was 24% and for SCLC 6% was reported. This poor survival rate is due to the delay in diagnosis resulting from the lack of early detection strategies for lung cancer [464]. So, there is a necessity for identifying biomarkers for prognosis and early diagnosis of this disease. Research from recent decades has shown that proteomics studies can identify biomarkers for lung cancer [465]. Proteins play a functional role in disease progression and they can be detected as diagnostic, prognostic, or treatment response biomarkers to lung cancer by using various techniques of proteomics. The following table summarized (Table 5) the various proteomics approaches used to detect the prognostic and diagnostic biomarkers of lung cancer.

### 4.6. Breast Cancer

Breast cancer is one of the primary leading causes of cancer deaths in women worldwide [473,474]. This has made breast cancer the most prevalent cancer globally. Although the mortality rates have declined recently because of early diagnosis and effective treatment regimens, some types of breast cancers have poor prognoses, mostly with metastatic tumors [475,476,477]. Breast cancer can be classified based on the specific proteins associated with the cell functions including receptors for estrogen (ER), progesterone (PR), and human epidermal growth factor receptor-2 (HER2) [478]. Around 15% of breast cancer tumors are classified as triple-negative breast cancer (TNBC) because of the lack of expression of ER, PR, and HER-2 receptors [478,479,480]. Among various forms of breast cancer, TNBC is very aggressive in nature and lacks hormonal response to ER, PR, and HER2 receptor-targeted drug therapies [480,481]. This makes the prognosis of TNBC very poor.

Over the past 20 years, advances in proteomics have allowed us to catalog, visualize, compare, and dissect patterns of proteoforms and epigenetic alterations in different forms of breast cancer tissues. These studies identify and provide insight into key drivers of oncogenic signaling, novel treatment strategies including response to therapies, and specific tumor characteristics. The table below summarizes (Table 6) the historical and recent advances in proteome-wide analysis in breast cancer to understand tumor biology, as well as the clinical applicability of these discoveries.

## 5. Proteomics Contribution to the Clinical Treatment of Cancer

Conventionally, proteins are identified in the clinic using antibody-based techniques, such as ELISA and IHC. However, these techniques require antibodies with higher affinity and specificity that could be low throughput and more expensive. Proteomics technologies with standardized workflow open a new possibility in clinical settings because of their low cost, high specificity, and high multiplex potential. There are several studies that demonstrate the importance of the proteomics approach for understanding cancer development. For instance, proteomics studies are widely used in clinical breast tumor samples. Rezaul et al. tested the protein expression profiles associated with ER status of breast cancer and found 236 differentially expressed proteins between ER-positive and ER-negative tumors [494]. Similarly, Gamez-Pozoet et al. reported the identification of more than 1600 proteins in triple-negative breast cancer samples [495]. Cha et al. found 298 significantly differentially expressed proteins when compared to the normal and ER-positive breast epithelial samples [496]. With the improvement of sample preparation and MS technologies, the quality of proteomic data also significantly improved. Liu et al. used LCM LC MS/MS and obtained data for more than 3500 proteins, of which 11 proteins were significantly changed in the patients [488]. With the use of the same technique, De Marchi et al. obtained the signature for four proteins that predict tamoxifen susceptibility in recurrent breast cancer [497]. Do et al. showed MRM-MS yielded more accurate HER2 expression levels compared to IHC in 210 breast cancer tissue samples in a clinical setup [498].

For leukemia patients, Xu et al. developed a proteomic classification system by analyzing 151 de novo acute leukemia patients using SELDI-TOF MS [499]. The proteomic profile of these patients correlated with the sub-type of leukemia, such as granulocytic AML, acute promyelocytic leukemia (APL), acute lymphocytic leukemia (ALL), and acute monocytic leukemia (AML). This proteomic classification suggested by Xu et al. holds promise to identify potential protein biomarkers for each subtype of acute leukemia. Aivado et al. used MS in the serum of leukemia patients to differentiate between pre-malignant myelodysplastic syndrome and malignant AML [500]. They found a decreased level of CXC chemokine ligands 4 and 7 in advanced MDS. Similarly, Braoudaki et al. used MALDI-TOF MS to identify the diagnostic biomarkers MOES, EZRI, and apoptosis inducing factor mitochondria associated 1 (AIFM1) to distinguish MDS and AML [501]. Apart from MS, RPPA is used to identify PTMs in the sample of leukemia patients. Kornblau et al. have used RPPA for 256 adult AML patients to show a distinct protein profile between myeloid (M0-M2) and monocytic (M4-M5) AML [502]. Similarly, the protein profile of patients in M0-M5 was different from patients in M6 and M7 categories. RPPA-based proteomic profiling has been developed to distinguish between different leukemia subtypes. Hoff et al. found different protein expression signatures between AML and APL and between AML and ALL [503,504]. RPPA has also been used to detect the prognostic biomarkers in leukemia. Quintas-Cardama et al. used RPPA to report a lower level of TRIM 62 in AML, which acts as a tumor suppressor [505]. In the same population, high phosphorylation of Serin 318–321 on forkhead box O-3 (FOXO3) and low expression of ASH2-like, histone lysine methyltransferase complex (ASH2L) were discovered as prognostic biomarkers [506,507]. Lectin galactoside-binding soluble 3 (LGALS3), transglutaminase 2 (TGM2), and Fli-1 proto-oncogene (FLI1) were discovered as individual prognostic biomarkers in AML using proteomic techniques [508,509,510].

In HCC, proteomics techniques have been widely used to investigate chemoresistant hepatic cancer. The chemotherapeutic agent 5-Fluorouracil (5-Fu) is used to treat HCC, but some patients develop chemoresistance against 5-Fu treatment. Liu et al. compared the proteome and phosphoproteome of the 5-Fu resistant Bel/5-Fu cell line with the parental Bel cell line using stable isotope dimethyl labeling combined with high-resolution MS [511]. They identified 8272 unique proteins and 22,095 phosphorylated sites. They found an increased phosphorylated level of PLCb3pS1105 in the Bel/5-Fu cell line along with the increased level of SRC and protein kinase C-d (PKCd) that involve PLCb3 phosphorylation in the Bel/5Fu cell line as compared with the parent Bel cell line. Chen et al. performed a comparative quantitative phospho-proteomics study to find the underlying mechanism of sorafenib resistance between the sorafenib-resistant HuH-7 cell line with its parental cell line [512]. A total of 1500 phosphoproteins were identified; of those, 533 were significantly upregulated in the resistant cell, including the AKT, mTOR, and FAK signaling pathway activation. These results suggest targeting EPH receptor A2 (EphA2) by its suppression increases susceptibility to sorafenib resistance cells, which could help better management of advanced HCC. Melas et al. investigated the change in phosphoproteomics level in three HCC cell lines upon treatment with eight drugs (gefitinib, sorafenib, vandetanib, bortezomib, dasatinib, lapatinib, erlotinib, and sunitinib) in order to identify the phosphoproteomic signatures for predictive drug efficacy [513]. Their results showed inhibition of AKT under TGF and AKT under HER are indicative of clinically failed drugs, while extracellular signal-regulated kinase 1 (ERK12) under the hepatocyte growth factor (HGF) is an effective drug.

In lung cancer, proteomics-based LC-MS analysis in the sputum of lung cancer patients provides important information for the diagnosis and management of lung cancer. Yu et al. used the LC-MS for the clinical diagnosis of lung cancer patients using sputum samples from lung cancer patients and healthy controls [514]. They found the level of five proteins (Enolase 1 (ENO1), DNAX activation protein 10 (DAP10), hemopexin, and a tumor-cleared protein related to low-density lipoprotein receptor, and nucleotide exchange factor guanine) were higher in sputum of cancer patients as compared with control. This also confirmed ENO1 as the first major early-stage lung cancer biomarker. The table below summarizes (Table 7) the list of proteomic biomarkers used in cancer therapy.

## 6. Role of Proteomics in Drug Discovery

A direction in the development of cancer therapeutics involves the incorporation of the proteomics signature of patients to develop a treatment plan. Current clinical treatment focuses on targeted therapy, such as selectively inhibiting the molecular drivers of cancer in specific patients. The overall goal is to reduce the suffering of patients from the side effects of the cancer drug and increase the efficacy of cancer treatments. Because of the heterogeneous nature of all cancers, targeted therapy is considered the most effective therapeutic strategy in cancer treatment. Cancers frequently develop chemotherapeutic resistance. Therefore, in order to accelerate the efficacious cancer treatment, new drug targets must be discovered.

Detection of interaction networks in cancer seems to be one of the potential approaches to searching for new targets for drug discovery. Protein kinases involved in numerous protein–protein interactions are major drug targets. However, detecting the protein–protein interaction in the cell is a difficult task knowing that there are 2–3 orders of higher magnitude of protein–protein interactions than the number of kinases reported. Several compounds that disrupt the protein–protein interactions are either in clinical trials for cancer therapies or approved by the FDA. For example, the B-cell lymphoma 2 (Bcl-2) family inhibitors disrupt the bcl-2/bh3 domain protein–protein interaction, which is involved in the increased malignancies. Oltersdorf et al. reported the Bcl-xL inhibitor ABT-737 in 2005 by using the proteomic approach [529]. The final optimization of ABT-737 into ABT-199 was approved for the treatment of chronic lymphocytic leukemia (CLL) by the FDA in 2016 [530,531]. Similarly, Zak et al. solved the co-crystal structure of the extracellular domains of programmed cell death protein 1 (PD-1) and programmed cell death ligand 1 (PD-L1) in 2015, which revealed that their flat interaction domain was difficult to target with small molecule [532]. Most of the drugs approved by the FDA for targeting PD-1/PD-L1 belong to the antibody group (e.g., pembrolizumab, nivolumab, atezolizumab, avelumab, and durvalumab) that fall into a class of immuno-oncology treatments of cancer and are known as checkpoint inhibitors [533].

Chromatin remodeling is a process that relaxes the control of gene expression in the cancer cell. The bromodomain (e.g., BET) is the most well-studied protein domain by protein structure analysis that recognizes the poly-acetylated histone tail. Currently, BET bromodomain-acetylated histone inhibitors are under clinical trials. Small molecule bromodomain inhibitors were effective on the solid tumor of the breast, liver, lung, prostate, brain, intestine, and pancreas [534,535]. Further structural studies of the P53-MDM2 complex have helped to develop small molecules to inhibit this interaction, as P53 loss of function contributes to 50% of human cancers [536,537]. Cancer cells resist apoptosis by upregulation of the inhibitor of apoptosis protein (IAP) or by mutation of the BCL-2 family member BAK and BAX that activate the intrinsic apoptotic pathway [538]. The interaction of the second mitochondrial-derived activator of caspases (SMAC) with that IAP prevents apoptosis. The structural studies led to the development of many small molecule inhibitors of the SMAC/IAP interaction in the cancer cell, which are currently in clinical phase trials [538].

Ferruci et al. showed the involvement of the hepatocyte growth factor (HGF)/HGF receptor (cMET) pathway in myeloma progression [539]. They used 2-DE coupled MS to show that the treatment of SU11274, a cMET inhibitor along with anti-myeloma drugs (Bortezomib and Lenalidomide) downregulates the levels of angiogenic proteins such as annexin A4 (ANXA4) and prohibitin (PHB), peroxiredoxin-6 (PRDX6), and annexin A2 (ANXA2), while it upregulates the level of calpain small subunit 1 (CPNS1). Armstrong et al. used proteomics and transcriptomics combinatorial approaches to investigate the drug action of AUY922 (a next-generation Hsp90 inhibitor) in prostate cancer explants of prostate cancer patients [540,541]. They found that interference of fibronectin, a cytoskeletal and ECM-related protein, by AUY922 decreases the invasive potential of prostate cancer cell lines. Roolf et al. investigated the effect of fms related receptor tyrosine kinase (FLT3) inhibitor sorafenib on acute leukemia cell lines (an FLT3 wild-type and mutated cell line) [541]. They used the phosphoproteomics approach to investigate the mechanisms underlying sensitivity to sorafenib in both FLT3 wild-type and mutated cell lines. They found that the MEK/ERK signaling pathway was affected in the FLT3 wild-type cell line, while the mTOR pathway was suppressed in the FLT3 mutated cell line. Further, Tripathi et al. have studied the chemotherapeutic resistance in small-cell lung cancer (SCLC) cell lines using LC-MS/MS analysis [542]. They found five cell surface receptors (EGFR, JAG1, iTGB1, EPHA2, MCAM) that exhibit significantly higher expression in chemoresistant, as compared to chemosensitive cell lines.

Chemical proteomics platforms with activity-based protein profiling (ABPP) in combination with MS are the most common technique to discover selective and in vivo active inhibitors for enzymes in the process of drug discovery [543]. Kinobead technology in combination with MS analysis is used to select a broad range of kinase inhibitors, which are immobilized to beads for the purification of kinases from cancer tissue samples or cells [544]. Further, chemical proteomics was established as a platform for fragment-based ligand discovery to map the interactions of small molecules and proteins in human cells [545]. Gefitinib, an inhibitor of the tyrosine kinase receptor and the epidermal growth factor receptor (EGFR), is used for the clinical treatment of NSCLC with EGFR mutation. To identify the kinase inhibitor resistance mechanisms in cancer cells, a chemical proteomic approach comprising kinobeads and quantitative MS was used to determine that the resistance resulted from the higher expression of Ephrin type-A receptor 2 (EPHA2) in cancer cells. Treatment with a multikinase inhibitor, dasatinib, and gefitinib, restored the chemotherapeutic sensitivity of gefitinib [546,547].

## 7. Discussion and Perspective

The ongoing advancements in proteomics technologies provide hope to meet the challenge of human proteome analysis, studded with the inherent complexity and the wide dynamic structural and functional range of proteins. The PTMs add enormous diversity to the proteome via the covalent attachment of chemical functional groups, such as phosphate, ubiquitin or alkyl groups, carbohydrates, lipids, and proteolytic cleavage, generating a plethora of proteoform. PTMs influence both physiological and pathological processes. To understand biological processes and discover novel biomarkers of diseases using proteomics, alterations in proteoforms and their expression levels need to be accessed [548]. In the case of oncoproteomics, the clinical and cancer heterogeneity adds further complexity. The protein profiles may differ between patients, which complicates the differentiation of candidate biomarkers from false results due to random variation. The complexity of samples, such as serum, urine, or tissues, and the wide dynamic range of protein concentrations make biomarker discovery difficult. The innovation in analytical approaches has drastically improved the analysis of complex biological mixtures, quantification of proteoforms, detection of low abundance proteoforms, analysis of protein complexes, and high-throughput applications. However, to identify relevant protein alterations accurate tools for sample selection, standardized methods of sample collection and preparation, enrichment methods (for low abundance components while depleting the most abundant representatives), state-of-the-art analytical methods, data processing, and data interpretation are required. This will improve reproducibility and generalizability.

A wide range of proteomic approaches are now available, ranging from the conventional to contemporary techniques, including the 2-DE, DIGE, high-resolution MS, iCAT, iTRAQ, SILAC, TMT, protein microarrays, protein domain microarray, protein aptamers, and protein terminomics, to name a few. The usage depends on the specific requirement and the availability. For example, the protein microarrays including antibody/antigen arrays provide a highly sensitive, high-throughput semi-quantitative or quantitative analysis for knowledge-based examinations. RPPA is a robust tool for protein biomarker validation in various cancers due to its minimal inter-assay variation [267,268,269,270], and is successfully used for large-scale patient profiling and diagnosis in various cancers [59,271]. Antibody arrays are resorted for proteomic profiling of low abundant proteoforms and are widely applied in high-throughput analysis of cancer biomarkers [273,274]. The antigen microarrays or functional protein arrays are used to investigate the interaction with proteoforms, lipids, small molecules, nucleic acids, and antibodies. Serological autoantibodies (AAbs) for cancer biomarker profiling have been identified using high-plex protein arrays in ovarian, gastric, bladder, prostate, and breast cancers [275,276]. Further, the use of nanoproteomics and single-cell proteomics has enhanced the chance to discover novel biomarkers, even with small sample sizes. However, small sample sizes may also result in strong reporting bias of a statistically significant proteomic biomarker and reduces the chance of detecting a true effect. Therefore, caution must be exercised when using small sample sizes.

Recently, various proteomics approaches have been developed; however, MS still plays a key role in the detection and characterization of proteoforms [549]. Top-down proteomics plays an important role in the analysis of intact proteins. It provides higher sequence coverage of target proteins and thus better characterization of proteoforms, sequence variations, unravelling disease mechanisms, and discovering new biomarkers [550,551]. However, the efficient front-end separation of intact proteins is difficult compared to the separation of peptide mixtures. The relatively recent middle-down approach has the potential for successful applications in proteomics, as well as the study of isolated/purified proteins [552]. The middle-down proteomics analyzes digested peptides obtained by enzymatic or chemical digestion instead of intact proteins; however, the size of the resulting peptides is greater than the ones that are usually encountered in the bottom-up approach. A relatively fewer number of peptides reduces sample complexity and enhances the probability of detecting more unique peptides, particularly those of greater length. Moreover, the enhancement in the sequence coverage of the proteins results in the detection of more PTMs compared to the bottom-up approach. The bottom-up approach plays a role in the large-scale analysis of high-complexity samples. The protein or protein mixtures are digested enzymatically or chemically prior to MS analysis to obtain a characteristic peptide fragment. It provides higher sensitivity and a better front-end separation of peptides compared with proteins; however, the limited protein sequence coverage of identified peptides results in loss of labile PTMs and ambiguity of the origin because of redundant peptide sequences [26]. Proteomic technologies are rapidly increasing in throughput, with advanced methods that allow for hundreds of proteomes to be recorded per day on a single mass spectrometer with high resolution, speed, selectivity, and sensitivity.

The advances in the ability to detect complex proteoforms and increased proteome coverage, along with better data compilation and databases, are facilitating superior data analysis for improved functional annotation of proteoforms. It can be utilized to compare protein profiles during physiological and pathological states, and access the changes in protein profiles and protein distribution in tissues and body fluids during cancer progression, thereby enhancing our understanding of the underlying mechanisms of tumorigenesis, which is vital for the development of more efficacious and less harmful treatments that can directly target altered proteins and deregulated pathways. Currently, proteomics is not only applied in the prognosis, diagnosis, and treatment but it is also employed to identify the characteristics of drug-resistant cancer cells and discover targets that can overcome drug resistance developed during anti-cancer treatment [553]. The cells that are resistant to anti-cancer agents in cancer exhibit specific proteoforms and molecular mechanisms, and are linked with the poor survival rate of patients [17,554,555]. These investigations may provide the possibility to modulate key proteins involved in drug resistance to maximize the reach of cancer therapeutics. The drug-resistant cancer cell attributes are related to stemness in development, progression, recurrence, and metastasis [556]. The proteomic analysis of cancer stem cells from drug-resistant cancer cell lines suggest new specific markers and therapeutic targets [493,557].

From a clinical translational point of view, there is a need to implement novel protein-based biomarkers under appropriate clinical settings, as the currently used cancer biomarkers are mostly for diagnostic purposes. The technological aspects of proteomics need further development to enhance detection accuracy, particularly at the early discovery stage. This can be achieved by increasing the detection resolution, standardizing workflows, using high-quality antibody probes, and further improving MS ion injection efficiency, detector sensitivity, and cycling speed. These advances will increase selectivity and sensitivity, and help to detect organ-specific biomarkers which are present at ultralow abundance. The combined power of proteomics technologies to validate biomarker candidates can be utilized to complement and balance the advantages and disadvantages of individual technologies. For instance, combinations of super-SILAC with LC-MS/MS or MS or aptamers with PEA were used for cancer biomarker discovery [420,520,558]. The cancer biomarker discovery has risen to a new horizon using single-cell and nanoproteomics. MS with the aid of a flow cytometry cell sorter and high-resolution trapped ion mobility spectrometry (TIMS)-TOF is being utilized in single-cell proteomics [559]. Especially in cancer immunotherapy, the surface protein phenotypes and single-cell proteome are serving as flashpoints for new biomarkers profiling [560]. The implementation of an integrated multi-omics approach, which encompasses the comprehensive and integrated analysis of combined data generated from various omics approaches, including proteomics, genomics, transcriptomics, metabolomics, and lipidomics, will enhance our understanding of disease biology [561] (Figure 3). Multiomics generates high dimensional large-scale datasets compared to a single analysis, which may provide invaluable information on cancer pathology and significantly contribute to the prognosis, diagnosis, and development of efficacious cancer treatments, and impact the care of cancer patients [561,562].

Because of the vast available literature on proteomics technologies some aspects, such as detailed PTM enrichment methods and sample preparation workflow, could not be covered in this review. However, to provide an overview of oncoproteomics techniques, we have presented the major and advanced proteomics technologies with exemplars of application in cancer. We were able to cover six different cancers although the proteomics application extends to all cancers.

## 8. Conclusions

In conclusion, the innovation in proteomics technologies that are now capable of detecting and tracking small protein alterations with high accuracy, reproducibility, and analytical throughput are aiding early cancer diagnosis, the discovery of novel potential cancer biomarkers, and improving the clinical treatment of cancer. The integrated approach of using proteomics with other omics data will further enhance our understanding of disease biology and potentially revolutionize clinical practice in the future.

## Figures and Tables

**Figure 1 proteomes-11-00002-f001:**
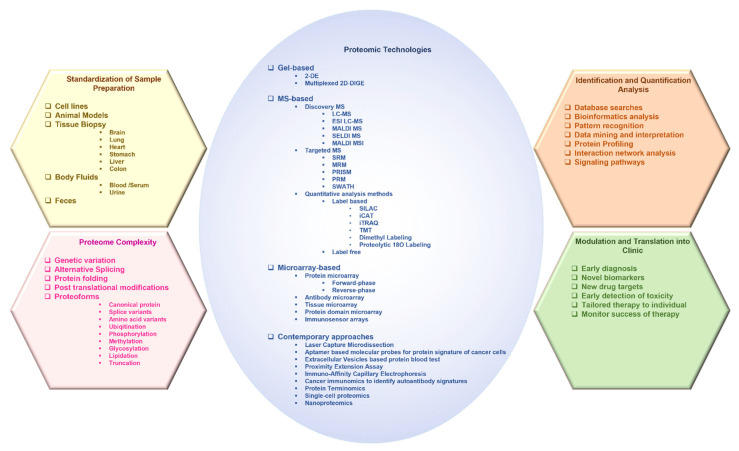
The various facets of proteomics investigations.

**Figure 2 proteomes-11-00002-f002:**
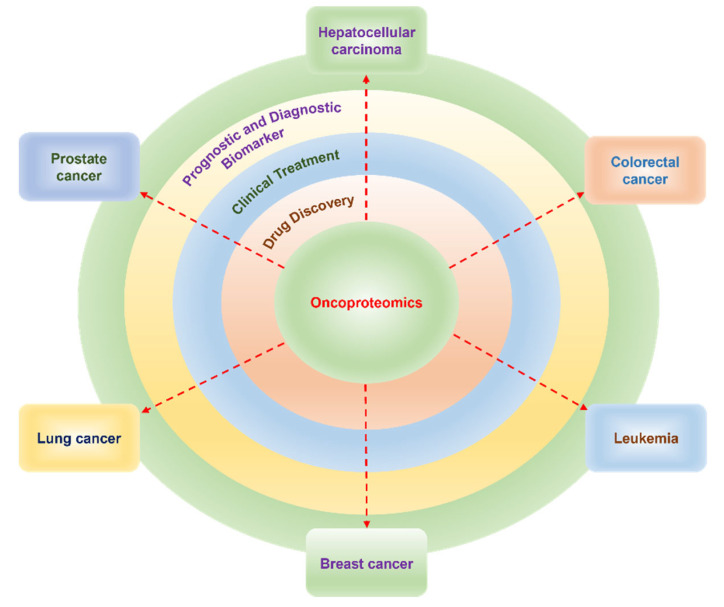
Role of oncoproteomics in drug discovery, prognostic and diagnostic biomarker development, and clinical treatment.

**Figure 3 proteomes-11-00002-f003:**
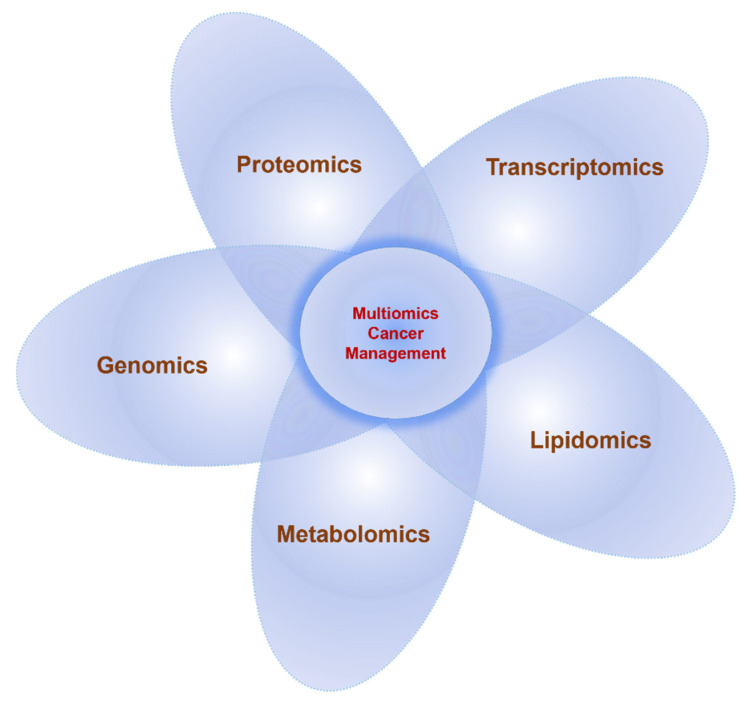
An integrated approach of using multiomics in translational research.

**Table 1 proteomes-11-00002-t001:** Studies on the diagnostic and prognostic relevance of proteomic biomarkers in hepatocellular carcinoma.

Reference	Proteomics Techniques	Biospecimen	Key Findings
Cao et al. [373]	Glycopeptide enrichment methods: hydrophilic affinity (HA) and hydrazide chemistry (HC) were used to complement LC-MS/MS	Human HCC cells	A total of 300 glycosylation sites within 194 glycoproteins were identified
Song et al. [374]	Pseudo triplex dimethyl labeling approach coupled with online RP-SCX-RP LC-MS/MS	Human HCC and normal liver tissues	A total of 1934 phosphopeptides from 1033 phosphoproteins were identified
Zhang et al. [375]	Lectin affinity chromatography (LAC)-nLC-ESI-MS/MS	Human HCC serum samples	A biomarker for postoperative recurrence of Quiescin Sulfhydryl Oxidase 1 (QSOX1) was identified
Jiang et al. [376]	A multi-parallel enrichment strategy based on the optimized ZIC-HILIC enrichment method assisted by a filter-coated 96-well plate MALDI-TOF MS	Three human HCC cell lines	A total of 5466 N-glycosites in 2383 glycoproteins were identified
Lin et al. [377]	Dimethyl labeling coupled with online 3DSCX-TiO2/RP LC-MS/MS and super-SILAC mix coupled with SIM/AIMS	Human HCC tissue	A total of 7868 phosphopeptides were identified
Block et al. [378]	LAC-2DE-HPLC-MS/MS	Animal models (woodchucks) of HCC	Golgi protein 73 (GP73) was identified as a diagnostic biomarker
Zhou et al. [379]	The 2-DE was followed by the fluorescence staining of glycoprotein and MALDI-TOF-MS/MS	Three human HCC cell lines	A total of 80 glycoproteins were identified
Chang et al. [380]	LC-MS/MS	Human HCC plasma samples	Indicators of HCC tumor grade C3 with mannan endo-1,4-beta-mannosidase (Man5), Man6, or Man7 glycoform at asparagine 85 were identified
Sun et al. [381]	Hydrazine chemistry and multiple protease digestion-dimethyl labeling-SCX-RP LC-MS/MS	Human HCC and healthy liver tissues	2329 N-glycosites on 1052N-glycoproteins were identified
Ang et al. [382]	LAC-2DE- MALDI-MS/MS	HCC patient serum samples	A diagnostic biomarker (haptoglobin) Hp was identified
Gao Q. et al. [383]	Nano-LC-MS/MS	Human tissue	Solute carrier family 10 members (1SLC10A1), pyrroline-5-carboxylate reductase 2 (PYCR2), and alcohol dehydrogenase 1A (Class I) (ADH1A) were identified

HCC—hepatocellular carcinoma, LC-MS/MS—liquid chromatography with tandem mass spectrometry, RP-SCX-RP LC-MS/MS—reversed phase-strong cation exchange LC-MS/MS, ZIC-HILIC—zwitterionic hydrophilic interaction liquid chromatography, nLC-ESI-MS/MS—nanoflow liquid chromatography-electrospray ionization–tandem mass spectrometry, 2DE—2D gel electrophoresis; LC/MS—liquid chromatography–mass spectrometry; MS—mass spectrometry; MALDI—matrix-assisted laser desorption/ionization; and TOF—time-of-flight.

**Table 2 proteomes-11-00002-t002:** Studies on the diagnostic and prognostic relevance of proteomic biomarkers in colorectal cancer.

Reference	Proteomics Techniques	Biospecimen	Key Findings
Kirana et al. [391]	2D-DIGE, MALDI-TOF MS	Fresh frozen tissue	Overexpression of cathepsin D (CTSD) in cells from the main tumor body showed a significant correlation with subsequent distant metastasis and shorter cancer-specific survival
Ku et al. [392]	TMT labeling, nano-LC-MS/MS	Fresh frozen tissue	Filamin A-interacting protein 1-like (FILIP1L) and plasminogen (PLG) upregulated in CRLM
Liu et al. [393]	TMT-labeling, LC-MS/MS	Fresh frozen tissue	fibronectin (FN1), metallo proteinase inhibitor 1 (TIMP1), thrombospondin-1 (THBS1), and periostin (POSTN) upregulated in CRLM
Shen et al. [394]	Acetylated peptide enrichment, TMT labeling, LC-MS/MS	Fresh frozen tissue	Acetylated histones, such as HIST2H3AK19Ac and H2BLK121Ac, changed while acetylated non-histones, such as tropomyosin beta chain (TPM2), K152Ac and alcohol dehydrogenase 1B (ADH1B), K331Ac altered in CRLM
van Huizen et al. [395]	Label-free nano-LC-MS/MS	Formalin-fixed paraffin-embedded tissue	Four collagen types, COL10A1, COL12A1 (the most abundant), COL14A1, and COL15A1 were upregulated in CRLM, while six non-collagen colon-specific proteins, cadherin-17 (CDH17), protein phosphatase-1 regulatory subunit-1B (PPP1R1B/DARP-32), keratin, type 1 cytoskeletal 20 (KRT20), carcinoembryonic antigen-related cell-adhesion molecule 5 (CEACAM5), cell-surface AA33 antigen (GPA33), and mucin-13 (MUC13), were upregulated in CRLM
van Huizen et al. [396]	Nano-LC-ESI-ETD-HCD	Formalin-fixed paraffin-embedded tissue	A lower ratio of 4xHyp at position 584 of the collagenalpha-2(I) chain (COL1A2) was found in CRLM
Fahrner M et al. [397]	Label-free LC-MS/MS	Formalin-fixed paraffin-embedded tissue	Metabolic proteins such as pyruvate carboxylase (PC) and fructose-bisphosphate aldolase B (ALDOB), and fructose-1,6-bisphosphatase 1 (FBP1) were upregulated in CRLM. Immune system proteins were enriched such as complement components C1, C4, C5, and C9 in CRLM. Structural proteins were depleted, such as desmin (DES), synemin (SYNM), and filamin-C (FLNC) in CRLM
Naba et al. [398]	ECM enrichment, off-gel electrophoresis, LC-MS/MS	Fresh frozen tissue	Hemopexin (HPX), osteopontin/secreted phosphoprotein 1 (SPP1), cartilage oligomeric matrix protein (COMP), insulin-like growth factor-binding protein complex acid labile subunit (IGFALS), fibronectin type III domain-containing protein1 (FNDC1), bone morphogenetic protein 1 (BMP1), and complement C1q tumor necrosis factor-related protein 5 (C1QTNF5). Extracellular matrix protein signatures are potential tissue or serological biomarkers
van Huizen et al. [399]	Label-free nano-LC-MS/MS	Formalin-fixed paraffin-embedded tissue	Hydroxylation of collagen was significantly lowered in CRLM and primary CRC as compared with a normal colon. Eleven peptides with a specific number of hydroxylation were downregulated in CRLM as compared with normal liver tissue
Kim et al. [400]	2-DE, MALDI-TOF MS	Fresh frozen tissue	Serpin family A member 1 (SERPINA1), apolipoprotein AI (APOA1), intelectin 1 (ITLN1), desmin (DES), diazepam-binding inhibitor (DBI), succinate dehydrogenase complex flavoprotein subunit A (SDHA), and carbonic anhydrase 1 (CA1) were upregulated in CRLM
Voß et al. [401]	Label-free LC-MS/MS	Fresh frozen tissue	Fifty-six extracellular matrix-associated proteins including tenascin C (TNC), nidogen-1 (NID1), fibulin-1 (FBLN1), and vitronectin (VTN) were upregulated
Yuzhalin et al. [402]	Extra Cellular Matrix enrichment, label-free, nano-LC-MS/MS	Fresh frozen tissue	Increased level of citrullinated proteins in CRLM as compared with normal liver peptidyl arginine deiminase 4 (PAD4)-driven citrullination of the extracellular matrix is essential for CRLM growth Other upregulated proteins include versican (VCAN), metalloproteinase inhibitor 1 precursor (T1MP1), latent-transforming growth factor beta-binding protein (LTBP) 1–3, epithelial discoidin domain-containing receptor 1 (DDR1), and protein S100-A10 (S100A10)
Yang et al. [403]	1D and 2-DE, nano-LC-MS/MS	Fresh frozen tissue	Olfactomedin 4 (OLFM4), CD11b/integrin alpha m (ITGAM), and integrin alpha-2 (ITGA2) significantly upregulated in primary CRC and CRLM
Kirana C et al. [404]	2-DE, MALDI-TOF MS	Fresh frozen tissue	HLA class I histocompatibility antigen B alpha chain (HLAB), A disintegrin, and metalloproteinase with thrombospondin motifs 2(ADAMTS2), latent-transforming growth factor beta-binding protein 3 (LTBP3), protein jagged-2 (JAG2), and nucleoside diphosphate kinase B (NME2) were upregulated in tumor cells and associated with CRC progression by invasion, metastasis, and CRC-specific survival
Michal et al. [405]	Label-free LC-MS/MS	Formalin-fixed paraffin-embedded tissue	Upregulation of matrix metalloproteinase 7 (MMP7) and dehydropeptidase 1 (DPEP1) in the poor-prognosis group. Downregulation of lysyl oxidase-like 1 (LOXL1) in the poor-prognosis group. A third of differentially expressed proteins were associated with the extracellular matrix
Turtoi et al. [406]	MALDI-MS imaging, nano-UPLC-qTOF MS	Formalin-fixed paraffin-embedded tissue	The latent-transforming growth factor beta-binding protein 2 (LTBP2) and transforming growth factor-beta-induced protein ig-h3 (TGFBI) were upregulated in CRLM and were absent in normal tissues
Yang et al. [407]	Label-free nano-LC-MS/MS	Fresh frozen tissue	Nine key proteins were identified in CRLM: heat shock protein family D member 1 (HSPD1), eukaryotic translation elongation factor 1 gamma, heterogeneous nuclear ribonucleoprotein A2/B1 (HNRNPA2B1), fibrinogen beta chain (FGB), Talin 1 (TLN 1), adaptor-related protein complex 2 subunit alpha-2 (AP2A2), serrated RNA effector molecule homolog (SRRT), apolipoproteinC3 (APOC3), and phosphoglucomutase 5 (PGM5). The fibrinogen α chain was reported as a key biomarker for CRLM
Chen et al. [408]	HPLC-MS/MS (orbitrap fusion)	Exosomes purified from the serum of CRC and normal patients	Identified metalloproteinase-9, galectin-3 binding protein, and the insulin-like growth factor
Shiromizu et al. [409]	LC-MS/MS (Q exactive)	Exosomes purified from the serum of CRC and normal patients	Identified mucin-5B, matrixmetalloproteinase-9, and transferrin receptor protein 1
Choi et al. [410]	LC-ESI-MS/MS (LTQ)	Microvesicles derived from CRC patient ascites	Identified the G-protein-coupled receptor E5, galectin-3, epithelial cell adhesion molecule, aminopeptidase N, and trophoblast glycoprotein

CRLM—colorectal liver metastasis, CRC—colorectal cancer, LC-MS/MS—Liquid chromatography with tandem mass spectrometry, LC-ESI-MS/MS—liquid chromatography-electrospray ionization–tandem mass spectrometry, 2-DE—2D polyacrylamide gel electrophoresis; LC/MS—liquid chromatography–mass spectrometry; MS—mass spectrometry; MALDI—matrix-assisted laser desorption/ionization; TOF—time-of-flight; HPLC—high performance liquid chromatography; and UPLC—ultra performance liquid chromatography.

**Table 3 proteomes-11-00002-t003:** Studies on the diagnostic and prognostic relevance of proteomic biomarkers in Leukemia.

Reference	Proteomics Techniques	Biospecimen	Key Findings
Nepstad et al. [420]	Super-SILAC DDA LC-MS/MS	AML cells derived from patients	Higher phosphorylation of transcription regulators decreased cytokine release and increased integrin expression on cells from acute myeloid leukemia (AML) patients with high constitutive activation of the PI3K-AKT-mTOR signaling pathway
Boer et al. [421]	Label-free DDA, LC-MS/MS	Peripheral blood and bone marrow cells from AML patients	Differential expression of leukemia-enriched plasma membrane proteins on distinct AML subclones. Some of the proteins (e.g., interleukin 3 receptor subunit alpha (IL3RA), IL2RA, T cell immunoglobulin and mucin-domain containing-3 (TIM3), and cluster of differentiation 44 (CD44), CD96, CD47, CD32, CD99, and CLEC12A have been previously identified by other non-MS-based technologies
Reikvam et al. [422]	Label-free DDA, LC-MS/MS	Leukemic cells from the peripheral blood of AML patients	Patient subsets with high constitutive cytokine release levels show high expression of proteins involved in intracellular signaling interacting with integrins, ras-related C3 botulinum toxin substrate 1 (RAC1), and spleen associated tyrosine kinase (SYK). AML cells with low cytokine release showed high expression of transcriptional regulators
Aasebo et al. [423]	Label-free DDA, LC-MS/MS	Circulating AML blast from the peripheral blood of AML patients	The constitutive release of mediators from primary AML differs from the intracellular protein levels
Tong et al. [419]	Label-free DDA, LC-MS/MS	Cell suspension AML patients and control	The study showed the connection between the protein tyrosine kinase and protein tyrosine phosphatase, and its effect on protein-phosphotyrosine signaling networks
Forthun et al. [416]	Label-free DDA, SRM, LC-MS/MS	Leukemic cells from AML patients	Phosphoprotein profiles revealed blast differentiation and cytogenic risk stratification
Grønningsæter et al. [424]	Label-free DDA, LC-MS/MS	AML cells derived from AML patients	Strong antiproliferative and proapoptotic effects of metabolic pathways inhibitors were observed on the cells of AML patients
Raffel et al. [425]	TMT DDA LC-MS/MS	AML cells from the bone marrow aspirations of AML patients	The expression of cell adhesion molecules, proteins of the oxidative phosphorylation process, and spliceosome factors were characterized in leukemia stem cells (LSCs)
Raffel et al. [426]	TMT DDA LC-MS/MS	Patient AML bone marrow, cord blood, and healthy mobilized peripheral blood samples	BCAA transaminase 1 (BCAT1) was enriched in leukemia stem cells (LSCs) and linked with a branched-chain amino acid (BCAA) metabolism to epigenomic and post-translational hypoxia-inducible factor 1-α (HIF1α) regulation via α-ketoglutarate (αKG)-dependent dioxygenase
Aesebo et al. [412]	Super-SILAC DDA LC-MS/MS	Primary cells from AML patients	High expression of RNA processing proteins, low expression of vacuolar-type ATPase (V-ATPase) proteins, and higher activity of casein kinase 2 (CSK2) and cyclin-dependent kinases (CDKs) could help predict chemo-resistant AML relapse
Brenner et al. [427]	Super-SILAC DDA LC-MS/MS	AML cells derived from the peripheral blood of patients	Transcription factors and proteins involved in mRNA splicing were highly expressed in AML cells with self-renewal capacity
Aesebo et al. [428]	Super-SILAC DDA LC-MS/MS	Primary cells from the peripheral blood of AML patients	Higher expression of mitochondrial ribosomal subunit proteins, RNA processing proteins, DNA repair proteins, and high activity of CDKs at AML relapse
Alanazi et al. [429]	iTRAQ DDA LC-MS/MS	Peripheral blood and bone marrow cells from AML patients	Over-expression of nuclear S100A4 in AML cells. Nuclear S100A4 is crucial for AML survival
Nepstad et al. [430]	Super-SILAC DDA LC-MS/MS	AML cells derived from patients	Enhanced phosphorylation and activation of PI3K-AKT-mTOR pathway by insulin was coupled to reduced antiproliferative effects of metabolic inhibitors in AML patient subsets
Schmidt et al. [431]	TMT DDA LC-MS/MS	Leukemic progenitors of AML patients	Protein modification and cytoskeleton reorganization proteins showed an altered abundance in the proteome of leukemic progenitor cells

LC-MS/MS—liquid chromatography with tandem mass spectrometry, nLC-ESI-MS/MS—nanoflow liquid chromatography-electrospray ionization–tandem mass spectrometry, 2DE—2D gel electrophoresis; 2D DIGE—2D differential in gel electrophoresis; LC/MS—liquid chromatography–Mass Spectrometry; MS—mass spectrometry; MALDI—matrix-assisted laser desorption/ionization; TOF—time-of-flight, SELDI—surface-enhanced laser desorption/ionization, iTRAQ—isobaric tags for relative and absolute quantitation; TMT—tandem mass tag; DDA—data-dependent acquisition; and SILAC—stable isotope labeling by amino acids in cell culture.

**Table 4 proteomes-11-00002-t004:** Studies on the diagnostic and prognostic relevance of proteomic biomarkers in prostate cancer.

Reference	Proteomic Techniques	Biospecimen	Key Findings
Itkonen et al. [437]	RPPA, O-GlcNAc chromatin consensus motif imposed by O-GlcNAc transferase (OGT) used as a bait; combination with MYC chromatin immunoprecipitation (ChIP)-MS	Prostate cancer cells	O-GlcNAc transferase (OGT) is an essential mediator in androgen-independency, which is the major mechanism of PCa progression
McCann et al. [438]	LC-MS/MS	Overexpression or depletion of ubiquitin specific peptidase 22 (USP22) in PCa cells and analysis of the ubiquitylome	Depletion of USP22 sensitizes cells to genotoxic insult; analysis of the USP22-sensitive ubiquitylome identified the nucleotide excision repair protein, xeroderma pigmentosum C (XPC), as a critical mediator of the USP22-mediated response to genotoxic insult
Drake et al. [439]	LC-MS/MS	Phosphoproteome of treated naïve and metastatic CRPC tissue samples integrated with genomic and transcriptomic data	Six major signaling pathways with phosphorylation of several key residues were significantly enriched in CRPC tumors; clinically relevant information (kinase target potential based on patient-specific networks) potentially suitable for patient stratification and targeted therapies in late-stage PCa is provided
Mariscal et al. [440]	LC-MS/MS	Palmitoyl proteome analysis of large and small cancer-derived PCa extracellular vesicles (EVs)	STEAP1, STEAP2 metalloreductase, and ABCC4 were identified as PCa-specific palmitoyl-proteins abundant in both EV populations; their localization in EVs was reduced upon inhibition of palmitoylation in the producing cells
Nguyen et al. [441]	LC-MS/MS	Human prostate cancer (PCa)-associated fibroblasts	(Phospho) proteomic profiling of PCa-associated fibroblasts-derived lysyl oxidase-like 2 (LOXL2) is an important mediator of intercellular communication within the prostate tumor microenvironment
Cui et al. [442]	Nano-LC-MS/MS	Proteomic experiments using a clickable palmitate probe (Alk-C16) between three individual pairs of androgen-treated and non-treated LNCaP cells	Androgen treatment significantly increased the palmitoylation level of eIF3L, which may be used as a biomarker for the diagnosis of early-stage PCa
Lee et al. [443]	MS	DU145 and RWPE1cells	Characterization of the ERG-regulated kinome. TNIK is suggested as a potential therapeutic target
Zhao et al. [444]	High-resolution MS/MS	Analysis of global phosphoproteomic changes induced by fish oil in human PCa	Pyruvate dehydrogenase α-1 is a target of omega-3 polyunsaturated fatty acids in human PCa
Faltermeier et al. [445]	MS-based phosphoproteomics dataset	Phosphoproteomics data from a mouse model of PCa progression. Integrated with gene expression analysis and literature mining	A total of 125 wild-type kinases implicated in human PCa metastasis were selected for screening for in vivo metastatic ability; the RAF family, MERTK, and NTRK2 kinases drive PCa bone and visceral metastasis, and are highly expressed in human metastatic PCa tissues, potentially representing important therapeutic targets
Wen et al. [446]	SILAC-MS	Quantitative proteomics to identify SUMOylated proteins in SUMO stably transfected PC-3 cells	More than 900 putative target proteins of SUMO were identified; mutation of newly identified SUMO modification sites of ubiquitin specific peptidase 39 (USP39) further promotes the proliferation-enhancing effect of USP39 on PCa cells
Jiang et al. [447]	LC MS/MS	Quantitative proteomic approach to compare protein phosphorylation in orthotopic xenograft tumors grown in either intact or castrated mice	Changes in phosphorylation of Yes1 associated transcriptional regulator (YAP1) and P21 (RAC1) activated kinase 2 (PAK2) and their elevated levels in CRPC were identified. YAP2 and PAK2 regulate cell colony formation and invasion in androgen-independent cells. PAK2 influences cell proliferation and mitotic timing. Pharmacologic inhibitors of PAK2 and YAP1 were able to inhibit the growth of androgen-independent PC-3 xenografts.
Toughiri et al. [448]	LC-MS/MS	Proteome analysis of Aurora-A substrates using small molecule inhibitor and reverse in-gel kinase assay in PC-3 cells	The nuclear mitotic apparatus (NuMA) becomes hypo-phosphorylated in vivo upon Aurora-A inhibition; mutation of three of these phospho-sites significantly diminishes cell proliferation and increases the rate of apoptosis.
Li et al. [449]	Nano LC-MS/MS	LNCaP cells were metabolically labeled with Alk-C16, a palmitate probe, and treated with R1881, an androgen, or DMSO, after which palmitoylome profiling was performed	Androgen treatment significantly increased the palmitoylation level of α-tubulin and Ras-related protein Rab-7a (Rab7a), which are essential for cell proliferation; in the supernatant of LNCaP cells, the palmitoylation level of α-tubulin was also increased following androgen treatment, which may represent a biomarker for early-stage PCa
Bai et al. [450]	MALDI-TOF-MS analysis	Proteomics analysis to determine the O-glycan profiles of PCa cells metastasized to bone (PC-3), brain (DU145), lymph node (LNCaP), and vertebra (VCaP) in comparison to immortalized RWPE-1 cells derived from normal prostatic tissue.	PCa cells exhibit an elevation of simple/short O-glycans, with a reduction of complex O-glycans, increased O-glycan sialylation, and decreased fucosylation. Core 1 sialylation is increased in all PCa cells. The expression of sialyl-3T antigen, which is the product of ST3Gal-I is increased. ST3Gal-I is associated with PC-3 cell proliferation, migration, and apoptosis. Downregulation of ST3Gal-I reduces the tumor size in the xenograft mouse model.
Clark et al. [451]	Nano-ESI-LC-MS/MS	EV-derived glycoproteins upon overexpression of FUT8 in PCa cells	A reduced number of vesicles secreted by PCa cells. Increase in the abundance of proteins associated with cell motility and PCa metastasis. Altered glycans on select EV-derived glycoproteins
Theurillat et al. [452]	SILAC-MS	Changes in the ubiquitin landscape induced by prostate cancer-associated mutations of speckle-type POZ protein (SPOP) in immortalized prostate epithelial cells expressing endogenous SPOP	DEK proto-oncogene and tripartite motif containing 24 (TRIM24) are effector substrates consistently upregulated by SPOP mutants with decreases in ubiquitination and proteasomal degradation resulting from heteromeric complexes of wild-type and mutant SPOP protein; DEK stabilization promotes prostate epithelial cell invasion
Drake et al. [453]	MS	Phosphotyrosine peptide enrichment and quantitative mass spectrometry (MS) in oncogene (non-tyrosin kinase)-driven mouse model of PCa progression	Elevated tyrosine kinase signaling (EGFR, EPHA2, JAK2, ABL1, and steroid receptor coactivator (SRC) tyrosine kinase activation) was observed
Li et al. [454]	LTQ Orbitrap LC-MS/MS	Cell surface Thomsen–Friedenreich (TF) antigen proteome profiling of metastatic PCa cells	A cluster of differentiation 44 (CD44), CD49f, CD133, CD59, CD138, EphA2, α2 integrin, β1 integrin, transferrin receptor, and profilin express TF antigen. TF antigen-positive prostate cancer cells form significantly more and larger prostaspheres under both non-differentiating and differentiating conditions and express higher levels of stem cell markers.
Ino et al. [455]	MS	Comparative phosphoproteome analysis of a PCa cell line LNCaP, and an LNCaP-derived androgen-independent cell line LNCaP-AI	The phosphorylation level of THRAP3 was significantly lowered in LNCaP-AI cells; the nonphosphorylatable mutant form of THRAP3 and the phosphorylation-mimic form differ significantly in protein binding repertoire; many of the differentially interacting proteins were identified as being involved in RNA splicing and processing
Gulati et al. [456]	SILAC-MS	Knockdown of E6-associated protein (E6AP) in DU145 cells and analysis of a proteome	Clusterin is a novel target of E6AP; the concomitant knockdown of clusterin and E6AP partially restores cell growth
Gao et al. [457]	LC-MS	Highly aggressive PC-3 and PC-3M cells	Compared phosphoproteomics of differentially expressed kinases. PAK2, STE20-like kinase (SLK), mammalian STE20-like protein kinase 4 (MST4), mitogen-activated protein kinase 2 (MAP2K2), and A-Raf proto-oncogene, serine/threonine kinase (ARAF) were kinases that were potentially associated with increased migration in PC-3M cells
Hoti et al. [458]	LC-MS/MS	Comprehensive proteomic approaches of alpha (1,6) fucosyltransferase (FUT8) overexpressing PCa cells	EGFR and its downstream signaling were upregulated; cell survival was increased in androgen-depleted conditions
Sharma et al. [459]	MS	Palmitoyl proteome analysis of breast, PCa cell lines and ±DHHC3 ablation	Putative substrates include 22–28 antioxidant/redox-regulatory proteins and ablation of protein acyltransferase DHHC3 elevated oxidative stress. DHHC3 ablation in combination with chemotherapeutic drug treatment elevated oxidative stress with a greater than additive effect, and enhanced the anti-growth effects of the chemotherapeutic agents. DHHC3 ablation synergized with poly-ADP ribose polymerase (PARP) inhibitor PJ-34, to decrease cell proliferation and increase oxidative stress
Hoti et al. [460]	iTRAQ MS and LC- MS\MS	Proteomics of androgen-dependent and androgen-resistant LAPC4 cells	Alpha (1,6) fucosyltransferase (FUT8) was significantly overexpressed in the androgen-resistant LAPC4 cells; an overexpression of FUT8 might be responsible for the decreased PSA expression in prostate cancer specimens
Lee et al. [461]	LC-MS/MS in combination with SILAC	Phosphoproteomics of metastatic docetaxel-resistant PCa cell lines (DU145-Rx and PC-3-Rx)	Increased phosphorylation of focal adhesion kinase (FAK) mediates chemoresistance in CRPC

O-GlcNAc—O-linked β-N-acetylglucosamine, PCa—prostate cancer, CRPC—castration-resistant prostate cancer, ABCC4—ATP binding cassette subfamily C member 4, eIF3L—eukaryotic translation initiation factor 3 subunit L, EGFR—epidermal growth factor receptor, EPHA2—ephrin type-A receptor 2, JK2—Janus kinase 2, ABL1—ABL proto-oncogene 1, THRAP 3—thyroid hormone receptor-associated protein 3, PAK2—P21 (RAC1) activated kinase 2, LC-MS-MS—liquid chromatography with tandem mass spectrometry, nano-LC-ESI-MS/MS—nanoflow liquid chromatography-electrospray ionization–tandem mass spectrometry, 2DE—2D gel electrophoresis; LC/MS—liquid chromatography–mass spectrometry; MS—mass spectrometry; MALDI—matrix-assisted laser desorption/ionization; TOF—time-of-flight, iTRAQ—isobaric tags for relative and absolute quantitation; SILAC—stable isotope labeling by amino acids in cell culture.

**Table 5 proteomes-11-00002-t005:** Studies on the diagnostic and prognostic relevance of proteomic biomarkers in lung cancer.

Reference	Proteomics Techniques	Biospecimen	Key Findings
An. et al. [466]	LC-ESI MS/MS	Serum of lung cancer patients	Thirty-two different proteins were identified
Geary et al. [467]	sequential windowed Acquisition of all theoretical fragment ion MS	Serum of lung cancer patients	Eleven different proteins were identified
Li. et al. [468]	iTRAQ-2DE-LC MS/MS	Plasma of lung cancer patients	Multiple inositol polyphosphate phosphatase 1, thyroxine-binding globulin, mannan-binding lectin serine protease 1, cathelicidin antimicrobial peptide, carnosine dipeptidase 1, fibrinogen-like protein 1, ADAMTS-like protein 4, and haptoglobin were identified
Sabrkhany et al. [469]	nLC-MS/MS	Plasma of lung cancer patients	Forty-nine different proteins were identified
Zhou et al. [470]	LC-MS/MS	Serum of NSCLC patients	Elongation factor 1, alpha 2, proteasome subunit alpha type, and spermatogenesis-associated protein were identified
Chae et al. [471]	VeriStrat test MALDI-TOF MS	Serum of NSCLC patients	The VS-Good group demonstrated significantly higher progression-free survival (PFS) and overall survival (OS) compared to the VS-Poor group among overall NSCLC patients, regardless of treatment
Muller et al. [472]	MS	Serum of patients with advanced NSCLC treated with nivolumab	A total of 274 MS protein signatures were associated with progression-free survival (PFS) and overall survival (OS) in patients

NSCLC—non-small cell lung cancer, nLC-ESI-MS/MS—nanoflow liquid chromatography-electrospray ionization–tandem mass spectrometry, 2DE—2D gel electrophoresis; LC/MS—liquid chromatography–mass spectrometry; MS—mass spectrometry; MALDI—matrix-assisted laser desorption/ionization; TOF—time-of-flight, iTRAQ—isobaric tags for relative and absolute quantitation.

**Table 6 proteomes-11-00002-t006:** Studies on the diagnostic and prognostic relevance of proteomic biomarkers in breast cancer.

Reference	Proteomic Techniques	Biospecimen	Key Findings
He et al. [482]	Label-free LC-MS/MS	Breast cancer tissue	Heat shock protein (HSP) 70 kDa-8, periostin, RhoA, actinin alpha 4, cathepsin D, preproprotein, annexin 1, and aldehyde dehydrogenase 1 family member A1 (ALDH1A1), G3BP stress aranule assembly factor 1 (G3BP) were upregulated and Thymosin-β4, transketolase, and transferrin were downregulated as prognostic biomarkers and drug targets
Campone et al. [483]	iTRAQ labeling MALDI-MS/MS	Breast cancer tissue	Desmoplakin (DP), thrombospondin-1 (TPS1), and tryptophanyl-tRNA synthetase (TrpRS) were upregulated as prognostic biomarkers or drug targets
Suman et al. [484]	iTRAQ labeling MALDI-MS/MS	Breast cancer tissue and serum	Alpha-2-macroglobulin (A2M) was upregulated, and complement component 4 binding protein alpha (C4BPA) was downregulated as a prognostic biomarker
Sun et al. [485]	Dimethyl labeling LC-MS/MS	Breast cancer cell lines	Protein tyrosine phosphatase non-receptor type 12 (PTPN12) was downregulated as a prognostic biomarker
Semaan et al. [486]	Label-free LC-LTQ/FT-ICR MS	Breast cancer tissue	Tripartite motif containing 28 (TRIM28), HSP90-alpha, heterogeneous nuclear ribonucleoprotein A1 (hnRNP A1), clathrin heavy chain (CLTC), and myosin-9, heparin binding growth factor (HDGF) phosphorylated and HSP90, Abl interactor 1 (AB1), PTRF1 isoform 1 of polymerase I and transcript release factor, AHNAK nucleoprotein, and SEPT2 dephosphorylated were identified as drug targets
Lawrence et al. [487]	iBAQ (absolute quantitation) LC-MS/MS	Cell lines and tumors	NF-κB was upregulated as a prognostic biomarker
Liu et al. [488]	Label-free nLC-MS/MS)	Breast cancer tissue	UMP-CMP kinase (CMPK1), apoptosis-inducing factor 1, mitochondrial (AIFM1), ferritin heavy chain (FTH1), echinoderm microtubule-associated protein-like 4 (EML4), neutral alpha glucosidase AB (GANAB), catenin alpha-1 (CTNNA1), AP-1 complex subunit gamma-1 (AP1G1), syntaxin-12 (STX12), AP-1 complex subunit mu-1 (AP1M1), and F-actin capping protein subunit beta (CAPZB) proteins were upregulated and C-1-tetrahydrofolate synthase cytoplasmic (MTHFD1) was downregulated as a prognostic biomarker
Mittal et al. [489]	Label-free quantification LC-MS/MS	Breast cancer cell lines	Enolase 1 (ENO1) was upregulated as a prognostic biomarker
Liu et al. [490]	Label free nLC-MS/MS	Breast cancer tissue	Ferritin heavy chain 1 (FTH1) was upregulated as a prognostic biomarker
Wu et al. [491]	SILAC MS	Breast cancer cell lines	AXL receptor tyrosine kinase was upregulated as a prognostic biomarker
Tyanova et al. [492]	Super-SILAC LC-MS/MS	Cell lines and breast cancer tumors	Minichromosome maintenance complex component 5 (MCM5), stathmin 1 (STMN1), glutaminase (GLS), RNA terminal phosphate cyclaselike 1 (RCL1), chromosome 9 open reading frame 114 (C9ORF114), and ENO1 were upregulated and anterior gradient 2 (AGR2), melanophilin (MLPH), HID1 domain-containing (HID1), centralized master bidders list (CMBL), and forkhead box A1 (FOXA1) were downregulated as prognostic biomarkers
Shenoy et al. [17]	In-solution digestion and LC -MS/MS	Patient’s tissue and breast cancer cell lines	Pyrroline-5-carboxylate reductase 1 (PYCR1) was identified as a biomarker
Koh et al. [493]	In-solution digestion and LC -MS/MS	Breast CSCs and breast cancer cell lines	CD66c was identified as a biomarker

LC-MS-MS—liquid chromatography with tandem mass spectrometry, nLC-MS/MS—nanoflow liquid chromatography–tandem mass spectrometry, MS—mass spectrometry; MALDI—matrix-assisted laser desorption/ionization; TOF—time-of-flight, iTRAQ—isobaric tags for relative and absolute quantitation; SILAC—stable isotope labeling by amino acids in cell culture, and GC—gas chromatography.

**Table 7 proteomes-11-00002-t007:** List of biomarkers used in cancer therapy.

Proteomic Biomarker	Biospecimen	Cancer	Reference
Alfa-Feto-Protein-L3	Serum	HCC	[515]
Epidermal growth factor receptor (EGFR)-T790M	Tumor tissue	Lung cancer	[516,517]
Cancer antigen 125 (CA125), prealbumin, apolipoprotein A1, beta-2-microglobulin, and transferrin	Serum	Ovarian cancer	[518]
Thrombospondin 1 (THBS1), bromodomain and WD repeat domain containing 3 (BRWD3), epidermal growth factor receptor (EGFR), and complement factor H related 3 (CFHR3)	Plasma	Breast cancer	[519]
MBL-associated serine protease 1 (MASP1), osteopontin (OPN), paraoxonase 3 (PON3), and transferrin receptor (TFRC)	Plasma	Colorectal cancer	[520,521]
sE-cadherin, TSR1 ribosome maturation factor (TSR1), serum amyloid A (SAA), kallikrein related peptidase 3 (KLK3)	Serum	Prostate cancer	[522,523]
Peptide profiling	Serum	Cervical cancer	[524,525]
Serum amyloid A2 (SAA2), kallikrein B1 (KLKB1), apolipoprotein A1 (APOA1), and cluster of differentiation-44 (CD44)	Serum	Multiple Myeloma	[526,527,528]

HCC—hepatocellular carcinoma, MS—mass spectrometry; MALDI—matrix-assisted laser desorption/ionization; TOF—time-of-flight, and iTRAQ—isobaric tags for relative and absolute quantitation.

## Data Availability

No new data were created or analyzed in this study. Data sharing is not applicable to this article.

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
