# Peer review of "Advancements in Oncoproteomics Technologies: Treading toward Translation into Clinical Practice"

_proteomes, 2023, doi:10.3390/proteomes11010002_

Round 1

Reviewer 1 Report

With their review manuscript the authors started a quite complex project. A search in Pubmed with the key words proteome and cancer results in 38,000 hits. The authors decided to cite 587 of them. The aim of a review should be to collect the data and by working them out new knowledge should be generated. I have the impression that the manuscript in the actual version is not satisfactorily overcoming the listing phase. The lists of proteomic applications for the different cancers are impressing, but the outcome of clinic applications seems to be not convincing. Despite mentioning the importance of proteoforms, the authors do not recognize the paradigm change from the protein expression concept to the protein speciation concept (Jungblut et al. Chem Central J 2008 and Schlüter et al. Chem Central J 2009). A consequence of the protein speciation concept is that top-down methods are necessary to detect the functional unit, the proteoform or protein species. Bottom-up methods principally cannot reach the protein species level, because they destroy by digestion the protein species and the identified peptides cannot be assigned to certain protein species, they represent a mixture of protein species.  Most of the proteomic studies today are bottom-up analyses, maybe a reason of the low applicability for diagnosis and therapy?

There are several major concerns:

-       The manuscript is too long for a Journal Review

-       The paradigm change from protein expression to protein speciation should be included with its consequences for the applications.

-       The structure of the manuscript is not concise: 2-DE is not the alternative to MS, it should be 2-DE-MS versus LC-MS. 2-DE-MS is also an MS-technology. Make the structure more concise and it should fit with Figure 1

-       Grammar is poor throughout the manuscript. In the detailed concerns, only some examples are shown.

-       Hela cells are a perfect model system for proteomics. They are missing completely.

-       Stable isotope dimethyl labeling is not mentioned in the chapter of labelling techniques

-       “etc.” found many times in the manuscript, which does not represent a good style.

-       References are mostly not referring to the original publications. There are many examples I recognized, but I did not check this for each of the more than 500 references: Examples: Proteoforms, protein species, 2-DE, SILAC, metabolic labeling. How did the authors reduce the large number of publications from Pubmed? If a review is cited this should be mentioned.

-       A list of all proteomic markers, which are already used for diagnosis or therapy in the clinic would be interesting.

Minor and detailed concerns:

-       P2 line 47: The term proteoform is now used instead of protein isoforms, protein species, and protein variants [3] . This sentence is misleading. The term isoform is clearly defined by the IUPAC restricting to genetically caused forms. The term protein species was clearly defined in Jungblut et al. Chem Cent J. 2008. Each chemical modification on a protein leads to a new protein species. Protein variants are not clearly defined. The term proteoform was created 5 years later than protein species by Smith and Kelleher (the authors reference Nr.4). The term proteoform is a gene-centric definition (s. Kelleher J of Proteomics 2016). The phenomenon of protein speciation was the first time defined in the article Jungblut et al. Chem Central J 2008 and worked out in Schlüter et al. Chem Central J 2009.

-       P2 line 82: Thus, increasing the understanding of cancer pathological mechanisms. This is not a complete sentence

-       P2 line 83: Additionally, the proteomics has been. Better: Additionally, proteomics has been

-       P3 line 100: The general pipeline of a proteomic analysis consists of protein extraction or purification from tissue or cell lysates, followed by protease (usually tryptic) digestion, … This is misleading. It fits only to the bottom-up approaches. In 2-DE-MS and Top-down MS the proteins are first separated, or directly analyzed by MS, respectively. Digestion of proteins destructs the protein species and information about protein species is destroyed by the bottom-up approach.

-       P3 line 138: The difference in gel-spots can be compared between disease versus normal controls to identify alterations in protein abundance or changes in the abundance of certain PTMs. This sentence is not clearly understandable. Which difference in gel spots? How can the abundances of PTM be identified? What does this tell us? I can imagine that the abundance of a proteoform may be a potential result. But only in top-down approaches.

-       P3 line 142: The differentially expressed proteins can be identified by direct side-by-side comparison … The protein expression terminology should be avoided, because it is misleading (s. Jungblut et al. Chem Central J 2008 and Schlüter et al. Chem Central J 2009).

-       P4 line 159: Furthermore, large-scale detection, identification, and quantification of the pro- teoforms can be achieved via stable isotope-labeled 2-DE coupled with high-sensitivity Liquid Chromatography-tandem Mass Spectrometry (LC-MS/MS) [50], [51].  These are wrong citations: The first publication combining high-resolution 2-DE with SILAC is Thiede et al. (your Ref53).

-       P4 line 181: …in vacuum that finally detected by the detector. ?

-       P11 line 514: The primary limitation of SILAC is the requirement for samples to metabolically active (i.e., undergoing active protein synthesis). This sentence is not understandable.

-       P11 line 546: The Isotope-Coded Affinity Tag (iCAT) is an in vitro isotopic labeling method used for quantitative proteomics by MS [189], [190]. These citations are not the original papers describing the method.

-       P11 line 568: Another limitation of ICAT is that only two labels available which results in frequent experimentation and high cost if multiple samples need to be analysed. “Sentences” like this one are found throughout the manuscript.

-       P12 line 570: In addition, ICAT can only identify a maximum of about 400 proteins, which is much less than the 2DGE method, and the peptides contain large labels, which make database searching more difficult, especially for short peptides [189], [193]. This sentence is misleading. The number of identifyable proteins depends on MS sensitivity and is surely higher with high-sensitive MS technology.

-       P12 line 579: … is a mass-tagging

-       P12 line 605: Apart from the above shortcomings, the iTRAQ technique has high sensitivity and ideally suited for comparing normal/diseased/drug-treated samples, … : is ideally suited…..

-       P12 line 659: why in the presence of protease?

-       P15 line 722: …profiling of number of large samples with the flexibility of multiple different comparisons [251], [252].  …profiling of a number of large….

-       P15 line 756: 2.2.7. Laser capture microdissection: This is not an MS method and does not fit in here. It is an Extraction method.

-       P16 line 805: The bait molecule can be a cell lysate, or subcellular protein fraction [276]. How can a cell lysate be a molecule?

-       P17 line 833: This method can practically characterize over a thousand proteins or modified proteoforms with minimal immunogenic cross-relativity induced from antibody reaction mixtures [286]. What is cross-relativity? In my opinion it is not possible, at least now, to produce an antibody specific for a certain proteoform. An antibody against a protein with a phosphorylation at position 55 detects all proteoforms with this modification, because this antibody does not discriminate between all of the other PTMs. Therefore, there may be hundreds or thousands of proteoforms binding to an antibody against this protein with P at position 55. Only if there are only two proteoforms in the sample, a completely PTM-free and one with only the phosphorylation at position 55, this will work. But this situation is quite improbable. It can be expected that each protein is multiply modified in a cell or even more in a tissue.

-       P17 line 876 : Bio-Plex systems use of … Sentence cannot be understood.

-       P18 line 880: … enables simultaneous proteomics analysis. Better: proteome analysis

-       P19 line 945: An immunosensor arrays are … Grammar

-       2.4 Microneedles: Does this chapter fit in this review?

-       P20 line 1011: …. that it is minimally invasive, cost-effective technologies, able to deliver small molecule, macromolecular drugs or nanoparticles to the tumor tissue in a safe and controlled manner. Grammar

-       P21 line 1068: …patients and compared to healthy donors …   as compared to

-       2.8. Immuno-Affinity Capillary Electrophoresis: This chapter is full of grammar mistakes and misinterpretations

-       P22 line 1114: reference is missing

-       P37 line 1327: proteomic classification system by analyzing 151 de novo acute leukemia patients using SELDI-TOF MS [533]. Reference 533 has to be completed.

Reviewer 2 Report

In this review, Punetha et. al, deeply discussed oncoproteomics. But some concerns need to be addressed.

1, Which kind of order did authors follow to write this review? Time order?

2, PTMs enrichment (such as phosphorylation, glycosylation, methylation and so on) and chemical proteomics should be discussed.

3, part 6, table 1-6, authors listed some studies about biomarkers discovered using proteomics techniques. But most of the studies were published 5-10 years ago. I think the most-recently publications are needed.

4, Please discuss the limitation. 

Round 2

Reviewer 2 Report

Authors revised the manuscripts. But please double check singular and plural nouns.

  •  

Author Response

Dear Reviewer,

Thank you for giving us the opportunity to submit a revised version of the manuscript entitled “Advancements in oncoproteomics technologies: treading towards translation into clinical practice” for publication in the Journal of ‘Proteomes’. We appreciate the time and effort that you dedicated to providing feedback on our manuscript. We have incorporated all the suggestions and edits are highlighted by track changes in the word document. Please refer to the revised manuscript file.

  1. 1. Authors revised the manuscripts. But please double check singular and plural nouns.

Author’s Response: Thank you for suggestions. We have modified and revised the required sentences throughout the manuscript.
